# Subcritical Water Extraction of Natural Products

**DOI:** 10.3390/molecules26134004

**Published:** 2021-06-30

**Authors:** Yan Cheng, Fumin Xue, Shuai Yu, Shichao Du, Yu Yang

**Affiliations:** 1School of Pharmaceutical Sciences, Qilu University of Technology (Former Shandong Academy of Sciences), Jinan 250353, China; yancheng2020@qlu.edu.cn (Y.C.); xuefumin@qlu.edu.cn (F.X.); yushuai@qlu.edu.cn (S.Y.); dushichao@tju.edu.cn (S.D.); 2Shandong Analysis and Test Centre, Qilu University of Technology (Former Shandong Academy of Sciences), Jinan 250353, China; 3Department of Chemistry, East Carolina University, Greenville, NC 27858, USA

**Keywords:** natural products, subcritical water extraction, alkaloids, glycosides, flavonoids, essential oils, quinones, terpenes, lignans, organic acids, polyphenolics, steroids, carbohydrates

## Abstract

Subcritical water refers to high-temperature and high-pressure water. A unique and useful characteristic of subcritical water is that its polarity can be dramatically decreased with increasing temperature. Therefore, subcritical water can behave similar to methanol or ethanol. This makes subcritical water a green extraction fluid used for a variety of organic species. This review focuses on the subcritical water extraction (SBWE) of natural products. The extracted materials include medicinal and seasoning herbs, vegetables, fruits, food by-products, algae, shrubs, tea leaves, grains, and seeds. A wide range of natural products such as alkaloids, carbohydrates, essential oil, flavonoids, glycosides, lignans, organic acids, polyphenolics, quinones, steroids, and terpenes have been extracted using subcritical water. Various SBWE systems and their advantages and drawbacks have also been discussed in this review. In addition, we have reviewed co-solvents including ethanol, methanol, salts, and ionic liquids used to assist SBWE. Other extraction techniques such as microwave and sonication combined with SBWE are also covered in this review. It is very clear that temperature has the most significant effect on SBWE efficiency, and thus, it can be optimized. The optimal temperature ranges from 130 to 240 °C for extracting the natural products mentioned above. This review can help readers learn more about the SBWE technology, especially for readers with an interest in the field of green extraction of natural products. The major advantage of SBWE of natural products is that water is nontoxic, and therefore, it is more suitable for the extraction of herbs, vegetables, and fruits. Another advantage is that no liquid waste disposal is required after SBWE. Compared with organic solvents, subcritical water not only has advantages in ecology, economy, and safety, but also its density, ion product, and dielectric constant can be adjusted by temperature. These tunable properties allow subcritical water to carry out class selective extractions such as extracting polar compounds at lower temperatures and less polar ingredients at higher temperatures. SBWE can mimic the traditional herbal decoction for preparing herbal medication and with higher extraction efficiency. Since SBWE employs high-temperature and high-pressure, great caution is needed for safe operation. Another challenge for application of SBWE is potential organic degradation under high temperature conditions. We highly recommend conducting analyte stability checks when carrying out SBWE. For analytes with poor SBWE efficiency, a small number of organic modifiers such as ethanol, surfactants, or ionic liquids may be added.

## 1. Introduction

For thousands of years, herbal medicine has played a vital role in treating diseases, especially in East Asia. The bioactive components in herbs and plants are the basis to prevention and treatment of many diseases [1]. Due to its relatively low side effects against chemically synthesized drugs, much attention is given to the extraction and separation of a wide range of bioactive compounds from herbs and plants. The 1000-year-old extraction process of the active pharmaceutical ingredients (APIs) from medicinal herbs is the traditional herbal decoction (THD) method. However, there are some defects in THD, such as a long extraction time and decomposition of active pharmaceutical ingredients. Methanol, ethanol, *n*-hexane, petroleum ether, diethyl ether, chloroform, ethyl acetate, and glycerol are often used as extraction solvents to increase the extraction efficiency and reduce extraction time. Obviously, these organic solvents are volatile, flammable, mostly toxic, and expensive. Thus, they are not safe extraction fluids for herbs, plants, fruits, vegetables, and the like.

Among the various new green extraction and separation technologies developed recently, SBWE is the most promising one. Subcritical water refers to the liquid water at temperature and pressure below its critical point (Tc = 374.15 °C, Pc = 22.1 MPa). The pressure of the subcritical water must be higher than the vapor pressure at a given temperature to keep water in the liquid state. With the increase of temperature, the physical-chemical properties of subcritical water change drastically. Its dielectric constant, viscosity, and surface tension all decrease steadily, and its diffusion coefficient is improved with increasing water temperature [2,3,4,5].

As shown in Figure 1, at 27 °C and 100 MPa water is a polar solvent, and its dielectric constant is 81.2. Fortunately, when the temperature is raised to 350 °C at 100 MPa, the dielectric constant of water decreases to around 20. In another words, water’s polarity changes from strong polar to much less polar [2]. For example, the dielectric constant of water at 200 °C, 250 °C, and 300 °C is equivalent to that of acetonitrile, methanol, and ethanol, respectively (Figure 1). Thus, even much less polar compounds can be extracted by subcritical water at high temperatures. Theoretically, based on its tunable polarity, subcritical water can extract many substances by adjusting extraction temperature and pressure conditions.

In the actual extraction processes, since pressure has minimal influence on the dielectric constant of water, water’s temperature is adjusted to control the dielectric constant of water in order to mimic various organic solvents. This unique characteristic of subcritical water allows water as the sole extraction fluid without any co-solvents such as acids, alkalis, catalysts, or organic solvents; this meets the principles of green chemical extraction since water is nontoxic.

Subcritical water applications can be found in the following areas: (1) reversed-phase liquid chromatography using subcritical water as the sole mobile phase-subcritical water chromatography [3]; (2) extraction of environmental samples, such as the determination of organic pollutants in solid wastes, soils, sediments, and atmospheric particles [5,6]; (3) hydrolysis, degradation, polymerization, and synthesis reactions using subcritical water as both a solvent or a reactant [7]; (4) environmental remediation such as cleaning contaminated sewages and soils, decomposing pollutants (pesticides, polycyclic aromatic hydrocarbons, and polychlorinated biphenyls) and explosives [8]; (5) extraction of active ingredients from medicinal and seasoning herbs, vegetables, fruits, and other plant related matrices [4,9].

This review focuses on SBWE of natural products in recent years. Principles, mechanisms, static and dynamic extraction modes, and factors affecting SBWE efficiency have been described by Gbashi and coworkers [10]. Since various research has focused on the extraction of flavonoids, carbohydrates, glycosides, organic acids, polyphenolics, alkaloids, essential oils, quinones, terpenes, lignans, and steroids [11,12], we provide a systematic and comprehensive overview on SBWE conditions, the function and activities of the active ingredients and the subcritical extracts, analysis methods, and comparison with other extraction methods for the above-mentioned natural products. Subcritical water can simultaneously extract several active ingredients from natural products. Separation, identification, and quantification of each natural product compound in the subcritical water extracts are achieved by liquid chromatography, gas chromatography, infrared spectrum, and/or mass spectrometry.

## 2. Sample Matrices Extracted by Subcritical Water

In general, the sample matrices extracted by subcritical water include the following four groups: Plants, food by-products, fungi, and marine algae. Based on the analysis and statistics of the references cited in this review, we generated Figure 2 to show the percentages of each type of sample matrices extracted by subcritical water. As shown in Figure 2, 62.3% sample matrices extracted by subcritical water come from plants, followed by food by-products with 29.0%, marine algae with 4.9%, and fungi with 3.8%.

For some medicinal herbs, their whole plants have pharmaceutical values and are prescribed in traditional herbal medicine, while only part or parts of other medicinal herbs are used in herbal medicine. The parts of medicinal herbs include root, stems, leaves, flowers, seeds, and fruits. For example, the whole plants of *Hedyotis diffusa* Willd. [13] and *Centella asiatica* L. [14] have medicinal values and the leaves, nodes, petioles, and roots of these two herbs were grounded before SBWE. Stems and leaves of ginseng [15], root of *Sophora flavescens* Ait. [16] and *Rheum tanguticum* [17] and flowers of chamomile ligulate [18] were extracted using subcritical water. We carefully sorted out the medicinal herbs, and the percentage of each medicinal part is illustrated in Figure 3. One can see from Figure 3 the leaves of medicinal herbs that are most widely investigated in SBWE. Although the peels, hulls, brans, barks, shells, epicarps, pericarps, sorghums of fruits and seeds are the byproducts, they also contain many APIs. For example, onion, one of the most frequently consumed vegetables, is known to have many health benefits because of its flavonoid contents. Some results demonstrated that the onion peel extracts produced by SBWE technique have great potential as a source for useful antioxidants [19,20]. Grains, seeds, corn, and fruits have also been investigated using SBWE [21]. Among all papers reviewed, only one article reported that the wood of *Aquilaria malaccensis* has useful medicinal applications, and is used in traditional medicine to treat pain, fever, rheumatism, and asthma [22].

### 2.1. Plants

Plants extracted by subcritical water mainly include herbs [15,23,24,25,26,27,28,29], vegetables with medicinal values [30,31], fruits [32,33,34,35,36], oilseed crops [37,38,39,40], shrubs, grains, tea leaves and beans [41,42,43,44,45,46]. These plants have been used in traditional medicine for thousands of years in some countries and have been proved to possess plentiful pharmacologically active components. For example, ginseng is a valuable Chinese medicine, which has been used in China, Korea, Japan, and Brazil. Ginseng has been reported to contain a variety of bioactive chemical compounds including terpenes, polyacetylenes, alkaloids, vitamins, minerals, phenolics, flavonoids, and triterpenes [15]. These active components possess antioxidant, anti-inflammatory, antidiabetic, antineoplastic, cardiovascular, immunoregulatory, and neuroregulatory activities [26]. Oregano is an herbaceous plant native to the Mediterranean regions [47,48]. Some healthy properties have been attributed to this plant, such as its powerful anti-bacteria and anti-fungal effects related to carvacrol and thymol compounds and some antioxidant activity.

### 2.2. Food By-Products

In essence, by-products are the wastes produced by fruit peels, tea filters, seed residues, vegetable peels, chestnut barks, cocoa shells, grain bran, lotus seed epicarp, and stems [49]. Many kinds of fruits have peels, such as bananas, oranges, and apples. Although we often throw them away, fruit peels may contain many APIs. Therefore, research on the by-products is worth doing. For example, orange peel is the main waste by-product of the juice extraction industry [50]. Nonetheless, orange peel is an attractive source of bioactive compounds, which include plenty of phenolic and flavonoid compounds.

### 2.3. Marine Algae

Marine algae studied in SBWE include microalgae, seaweeds, and *Haematococcus pluvialis* [51,52,53]. Algae are rich in saturated fatty acid, monounsaturated fatty acid, and polyunsaturated fatty acid, which is healthy for the cardiovascular system. Omega-3, which plays a significant role in the body’s inflammatory pathways and cell health, is especially used for cancer prevention and therapy. Using SBWE technology, algae could step up as one of the potential sources for future generation of omega-3.

### 2.4. Fungi

Some fungi are also responsible for some diseases in plants and animals, which also can be as vegetables, such as the edible-medicinal fungi, which include mushrooms (*Lentinus edodes* [54,55,56,57]; *Grifola frondosa* [58], *Sagittaria sagittifolia* L. [50,51,52,53,54,55,56,57,58,59,60,61]), and *Cordyceps militaris* (*C. militaris*) [62]. *C. militaris* is a type of precious edible-medicinal fungi widely distributed around the world. Extensive research demonstrated that extracts of *C. militaris* have multiple pharmacological actions, such as anti-inflammation, improvement of insulin resistance, and antioxidant activity.

## 3. SBWE Systems and Extraction Mechanism

At the beginning, supercritical water (T > 374 °C, P > 22.064 MPa) was widely involved in extraction of coal, oil sand bitumen [63,64], and oxidation [65]. Supercritical water requires harsh experimental conditions such as extremely high temperature and high pressure, which often bring severe corrosion on experimental equipment and connection tubing. Since 1990s, subcritical water has been gradually replacing supercritical water in the field of extraction because subcritical water requires relatively mild temperature and pressure. Thus, subcritical water has been extensively employed in the extraction of organic pollutants [66,67], and researchers have attempted to couple SBWE with liquid chromatography to reduce the analytical steps in the extraction and chromatographic analysis process [68,69,70]. To improve the extraction efficiency, ultrasonic-assisted SBWE [71] and microwave-assisted SBWE [72] have also been reported.

### 3.1. Modes of SBWE

There are mainly two SBWE modes, namely static extraction and dynamic extraction. Static extraction refers to an extraction method in which subcritical water and the sample to be extracted are added to an extraction vessel, as shown in Figure 4, and then heated to the desired temperature under a moderate pressure to keep water in the liquid state. After the set extraction time is reached, the extractant is collected for chromatographic analysis. This extraction process is similar to accelerated solvent extraction. The static extraction efficiency is normally lower than that of dynamic extraction.

Dynamic SBWE is a continuous extraction process, which means that after the samples are added to the extraction vessel, water is continuously fed into the extractor with a pump, and the extraction is carried out at a fixed temperature or in gradient temperature conditions. The pump can be set either under constant pressure mode or constant flow mode. Dynamic SBWE not only accelerates the mass transfer efficiency and shortens the extraction time, but also achieves selectively continuous extraction. However, dynamic SBWE may cause blockage to the SBWE system.

In many cases, both static and dynamic SBWE modes are used. SBWE operators normally conduct a period of static extraction, then followed by a dynamic extraction. Although a majority of the SBWE work is conducted without solid trapping of the extracted analytes (as shown in Figure 4, top), SBWE with a solid trap (Figure 4, bottom), which collects the extracted analytes can be easily coupled with liquid chromatography for analysis, as discussed further below.

### 3.2. Online and Offline Coupling of SBWE with Liquid Chromatography

SBWE system was coupled with high-performance liquid chromatography via solid-phase trapping [69]. The offline coupling of SBWE with HPLC was reported [70]. The solid trap was physically removed from the SBWE system after each extraction and connected to an HPLC system for analysis. However, this step is eliminated in the online SBWE-HPLC approach. After SBWE, the analyte trap stays in place while the HPLC analysis is carried out by simply switching valves to connect the analyte trap with the HPLC unit.

Offline coupling of SBWE with subcritical water chromatography (SBWC) has also been reported [70]. SBWC refers to reversed-phase liquid chromatography using subcritical water as the sole mobile phase component. As shown in Figure 5 (top), extracted analytes are collected in the sorbent trap during the SBWE step. After SBWE extraction, the sample trap is connected to a subcritical water chromatography system (Figure 5, bottom). The analytes collected in the sorbent trap during SBWE is thermally desorbed by heating the sample trap and then they are delivered into the SBWC system for analysis. The obvious advantage of SBWE-SBWC coupling is that organic solvents are eliminated in both extraction and chromatographic analysis steps. The challenge is, however, that the thermal desorption normally requires high temperature, which may potentially cause analyte degradation.

### 3.3. Mechanisms of Subcritical Water Extraction

The main SBWE extraction mechanism follows the “like dissolves like” rule. As mentioned in the Introduction section, the polarity of water can be tuned by temperature [2,3]. As shown in Figure 1, the dielectric constant of water is dramatically lowered (becoming less polar) with increasing temperature. This means that water’s polarity is tunable by changing temperature. Because of this, polar compounds can be efficiently extracted by water at “lower” temperatures while less polar analytes require higher temperatures to achieve a reasonable extraction efficiency. Therefore, the less polar the analyte, the higher the temperature required. In addition, larger and more complex molecules require higher temperatures. Pressure does not have a significant effect on SBWE efficiency as long as the pressure is high enough to keep water in the liquid state.

## 4. Compounds Extracted by Subcritical Water

### 4.1. Flavonoids

Flavonoids, also known as bioflavonoids, are widely found in plants and berries. They are important natural compounds in human diets. They have been used to prevent and treat cardiovascular diseases. In addition, they have strong antioxidant activities and antibacterial activities. When high-flavonoid apples were fed to healthy mice, decreases in some inflammation markers were reported [73].

Generally speaking, flavonoids belong to phenols. Since they are widely investigated, they can be presented separately from phenols. Flavonoids have the general structure of a 15-carbon skeleton by connecting two benzene rings with a heterocyclic ring. The basic nucleus is 2-phenylchromone. Flavonoid compounds are usually poorly soluble in ambient water and most organic solvents. Table 1 summarizes SBWE of flavonoids from plant materials. 

Ko and coworkers have investigated the relationship between flavonoid structure and SBWE. They found that flavonoids with an OH side chain were optimally extracted at lower temperatures than O-CH_3_ and H side chains. The optimal temperatures of the glycoside forms are lower than that of the less polar aglycones [30]. Turner et al. found that different glycosidic compounds may be converted by their respective aglycones in less than 10 min reaction time in water from onion waste [20]. Similar results were obtained by Nkurunziza et al. [81] and Zhang et al. [83] who investigated the extraction of four flavonoids from okara and from *Puerariae lobata*, respectively.

Various works have been published on the extraction efficiency and extracting activities by comparing SBWE with hot water extraction (HWE) [15], maceration [76], Soxhlet [50,81], ultrasonic assisted extraction (UAE) [93,103], microwave assisted extraction (MAE) [93,103], and reflux extraction [83]. All of them found that SBWE was more effective or at least equivalent to other extraction methods if only using one solvent. Đurović and coworkers [103] have selected water as an appropriate solvent for further extraction based on the extraction efficiency of six various solvents (water, acetone, 96% ethanol, methylene chloride, *n*-hexane, and ethyl acetate). SBWE has the highest total polyphenolics (TP) and flavonoids contents, and antioxidant activity. Cytotoxic activity and antimicrobial activities against Staphylococcus aureus were confirmed. Moreover, Marín et al. [99] has compared a possible better fit of pressurized liquid extraction for prenylflavonoids and three solvents (hexane, ethanol, and water) were used. They concluded that pressurized liquid extraction using ethanol as a solvent after using hexane is better than only using water for prenylflavonoids. When a sequential extraction was used, where solvents were used in an order of increasing polarity (first hexane, later ethanol, and finally water), yields did not improve. Different extraction methods have notable influences on the selectivity of these processes.

As shown in Table 1, the effect of extraction temperature and time have greater impacts on extraction efficiency than pressure and sample-extraction liquid ratio. The optimal extraction temperature and time are summarized in Figure 6. The optimal extraction temperature of flavonoids is between 150 and 200 °C and the range of extraction time is between 10 and 50 min.

### 4.2. Polyphenols

Polyphenols, also known as polyhydroxyphenols, are a structural class that is mainly natural, by the presence of more than one phenolic unit and being deprived of nitrogen-based functions. Many fruits, vegetables, herbs, tea leaves, nuts, and algae contain high levels of naturally occurring phenols. It has been reported that polyphenols can resist oxidation [31]. As shown in Table 2, extractions of polyphenols can be carried out either using a sole solvent such as water, methanol, ethanol or a mixture of solvents such as ethanol-water and methanol-water-formic acid.

Most experiments have proven that SBWE is superior to maceration, Soxhlet, and MAE [109,112,125], whereas Vladić et al. concluded that the lower temperatures of SBWE are more convenient for the valorization of pomegranate peel and MAE is more efficient than SBWE for the extraction of phenolics from pomegranate peel while obtaining a 5-hydroxymethylfurfural-free extract [109].

Procyanidins are a class of polyphenols found in fruits, vegetables, and grains with potent chemo preventive activities. Gao et al. [119] successfully extracted procyanidins with subcritical water from sorghum bran, lotus seed epicarp, lotus seed pod and sea buckthorn seed, respectively. The results showed that extracts by SBWE had better reducing power, and antioxidant ability and antiproliferative ability on human hepatoma G2 cells. Moreover, the antioxidant and antiproliferative activity were found to be positively correlated with polyphenol concentration. In addition, Gao et al. have found a similar result, as the antioxidant of the sea buckthorn seed residue extracts was highly correlated with the content of the polyphenols. They demonstrated that extraction time and the water to solid ratio were the major factors that affected the extraction yield of polyphenols and the 2,2′-azino-bis-3-ethylbenzthiazoline-6-sulphonic acid (ABTS) radical scavenging activity of the extracts, while temperature was expressed as a crucial factor.

Curcumin belongs to polyphenols and is unstable and poorly bioavailable. Kiamahalleh et al. [128] studied some parameters on SBWE extraction efficiency of curcumin. Optimum extraction conditions are a temperature of 140 °C and pressure of 10 MPa. Other works on SBWE of curcumin can be found in literature [29,129,130,131].

Pinto and his co-workers [110] have evaluated the optimal SBWE conditions of antioxidants and polyphenols from chestnut shells using response surface methodology (RSM). The optimal extraction conditions were determined by RSM as 220 °C/30 min. At 150 °C, TP and antioxidant activity was decreased, which may be caused by the degradation of the tannin and the formation of other reaction products. In addition, a concentration of under 0.1 µg/mL extracts was safe. The reuse of this chestnut shells by-product is beneficial to the profitability of agro-industry, and to the environmental and economic sustainability. The optimal extraction temperature and time for SBWE of polyphenols are depicted in Figure 7. One can see that the optimal extraction temperature is between 130 and 200 °C, and the time is between 15 and 35 min.

### 4.3. Organic Acids

In general, organic acids in natural products are widely distributed in the leaves, roots, and fruits of the plants. The synthetic organic acids through chemical synthesis, enzymatic catalysis, and microbial fermentation are not discussed in this review. Organic acids are mostly soluble in water or ethanol and exhibit acidic properties, but they are difficult to dissolve in other organic solvents. It is generally believed that aliphatic organic acids have no special biological activity, but some natural organic acids such as citric acid, malic acid, tartaric acid, ascorbic acid, etc. have antibacterial, anti-inflammatory, hypoglycemic, antioxidant, and immune regulation effects. Depending on the organic acid in free state or in salt form, the extraction solvents could be water, dilute alkaline solution, diethyl ether, petroleum ether and cyclohexane, and other lipophilic organic solvents. A summary of recent studies on SBWE of organic acids are shown in Table 3.

The use of SBWE was explored for the extraction of gallic acid, chlorogenic acid, caffeic acid, ferulic acid, vanillic acid, and coumaric acid from various matrices. Inevitably, some other active components such as phenolics [23], flavonoids [108], proteins [23], lipids, peptides, amino acids, and other organic compounds were often coextracted. Švarc-Gajić et al. [134] have used SBWE for the extraction of alcohols, organic acids, sugars, and other organic compounds from both sweet and sour cherry stems, finding the chemical compositions of the two samples similar. Harun et al. [51] reported lipid extraction with a relatively high content of eicosapentaenoic acid from *Nannochloropsis gaditana*, finding 237 °C and 14 min to be the optimum extraction conditions.

Chun et al. [52] explored the power of the ionic liquid (IL)-assisted SBWE method in obtaining different phenolic compounds from the brown seaweed Saccharina japonica and found the imidazolium-based IL 1-butyl-3-methylimidazolium tetrafluoroborate acted as a catalyst. Compared with SBWE, SBWE + IL provided a progressive enhancement in extracting phenolics. At 175 °C, the contents of gallic, chlorogenic, gentisic, protocatechuic, caffeic, and syringic acids in extracting products were approximately 1.18-, 4.68-, 4.66-, 7.67-, 5.12-, and 5.08-fold higher than in SBWE. However, *p*-hydroxybenzoic acid had a slight deficiency (0.85-fold at 175 °C) and vanillic acid easily decomposed in SBWE + ILs.

Ravber et al. [38] have applied SBWE to simultaneous removal of oil extracts and water-soluble extracts. The optimal extraction yield of oil was obtained at 130 °C and mass/sample of 1:20 g/mL after 30 min. Hydrolysis of ester and glycoside bonded antioxidants occurred, which produced oil with much higher antioxidant capacities than oil extracted using the Soxhlet (*n*-hexane) method. In addition, Abdelmoez and coworkers [39] have also compared the efficiency of water extracts and *n-*hexane extracts from cottonseed. The optimum oil extraction conditions are 270 °C, 30 min, and the particle size is less than 0.5 mm.

The optimal SBWE extraction temperature and time are shown in Figure 8. One can see that the optimal extraction temperature of organic acids is mainly between 130 and 240 °C, and the time is between 10 and 50 min.

### 4.4. Glycosides

Glycosides are compounds in which sugars or sugar derivatives are bound to another type of non-sugar substance (also called aglycones, ligands or substituents). Glycosides are linked by an O- N-, S-, or C-glycosidic bond between a sugar and a non-sugar component, which are widely found in the root, stems, leaves, flowers, and fruits of plants. Most glycosides are colored crystals, and generally a little bitter.

Glycosides extracted by SBWE have been proven to have antioxidant activities and tyrosinase inhibitory activity, as shown in Table 4 [36,144,145,146,147,148,149,150,151,152]. Gao et al. [144] has performed SBWE of phenolic compounds from pomegranate seed residues at 80–280 °C. The results showed that TP increased with the rise of extraction temperature from 80 °C to 220 °C and decreased from 220 °C to 280 °C. At 80–220 °C, the breakage of the bonds led to the increase of TP, however, a higher temperature caused the phenolics to degrade. In addition, they compared SBWE with leaching and UAE using water (room temperature) and organic solvents namely methanol, ethanol, and acetone. TP and antioxidant capacities of SBWE (120 °C) were not as high as organic solvents; however, with respect to the extraction time (2 h for leaching vs. 30 min for SBWE) and toxicity, subcritical water is more acceptable. Meng and Cheng [149] have studied 13 phenolic compounds and 20 inorganic elements of *Erigeron breviscapus*. They also have found similar results, as the glycosides are not stable at a high temperature and with a long extraction time. For example, scutellarein and apigenine are the aglycones of corresponding acutellarin and apigenin 7-glucuronide, when at high temperature glycosidic bonds become unstable and begin to decompose to its glycone and aglycone. Haznedaroglu et al. [147] have optimized the parameters such as temperature, extraction time, and flow rate. Temperature and extraction time were found as the most effective parameters for TP and total flavonoids while extraction time and flow rate for anthocyanin contents. In addition, temperature and time were the leading parameters for the effectiveness of extracts on tyrosinase inhibition.

The optimal extraction temperature and time for glycosides in SBWE are shown in Figure 7. The optimal extraction temperature of organic acids is between 100 and 160 °C, and the time is between 20 and 60 min.

### 4.5. Carbohydrates

Carbohydrates is a very common term that include sugars, starch, and cellulose, which are an important class of organic compounds widely distributed in nature. The saccharides are divided into four groups: monosaccharides, disaccharides, oligosaccharides, and polysaccharides. As shown in Table 5, carbohydrates extracted by SBWE possess antioxidants [153], antimitotic [154], and growth inhibitory effects [55].

Traditional extraction solvents often utilize alkaline solution [154], acidic solution [156], and water as extraction solvents [157]. Chun et al. [154] carried out SBWE from *Pseuderanthemum palatiferum* Radlk and compared it with a conventional technique using 0.1 M sodium hydroxide. Conjugates obtained from SBWE at 150 °C exhibited better anticoagulant activity than those found at 200 °C and were comparable to that of the conventional technique. Villamiel et al. [156] and Zhang et al. [159] used citric acid solution to adjust the pH to 3.0, 5.0, respectively. Compared to the conventional extraction with citric acid, SBWE can obtain higher pectin yield, higher galacturonic acid content and a higher degree of methyl esterification. Similarly, Chun et al. [158] and Goosen et al. [169] carried out SBWE and compared it with conventional extraction using 0.05 M HCl. They all found that SBWE efficiency was significantly higher compared to the conventional methods.

SBWE and other three innovative technologies UAE, MAE, and supercritical fluid extraction (SFE) has been compared or coupled. Duan et al. [56] have used UAE to successfully extract the chief ingredient polysaccharides from *Lentinus edodes*. A Box-Behnken design (BBD) was applied to optimize the UAE condition including extraction temperature, extraction time, liquid-to-solid ratio, and ultrasonic power. It was demonstrated that the UAE sample has better antioxidant activities compared with other extraction methods (SBWE, UAE). Morales et al. [57] have found UAE or SBWE (200 °C, 11.7 MPa) were more effective to obtain β-glucan enriched fractions from shiitake mushrooms. They revealed that a combination of UAE + SBWE extracts showed larger glucose levels and lower mannose and galactose residues than the other extractions, suggesting certain extraction specificity towards β-glucans. Moreover, the extracts obtained after combination of technologies partially retained their immunomodulatory properties, but they showed high hypocholesterolemic activities according to in vitro studies.

*Sagittaria sagittifolia* L. is a healthy food source and a restorative for the adjuvant therapy of tuberculosis, night blindness, pancreatitis, diabetes, tracheitis, and urinary tract infections. Ma and his team [35] have conducted continuous work on the characterization, antioxidant, and immunological activities of SBWE extracts from *S. sagittifolia* L. The extracts’ structural features were elucidated using high performance liquid chromatography (HPLC), gas chromatography (GC), scanning electron microscopy (SEM), infrared spectroscopy (IR), atomic force microscopy (AFM), zeta potential, and Congo red methods. Extracted polysaccharides exhibited stronger antioxidant activity in vitro and more potent immunomodulatory activity. Therefore, the polysaccharides extracted from *S. sagittifolia* L. with SBWE have the potential to be used as immunoreactive agent in medicine and functional foods. Plaza et al. have found the Maillard, caramelization and thermoxidation, and Folin reaction from microalgae, macroalgae, and rosemary samples. It is the first time the neoformation of antioxidants during SWE extraction of different natural products is verified [165]. Sometimes the high operation temperatures may generate new bioactive compounds, and SBWE is a very promising technique for obtaining bioactive compounds from natural products. As shown in Figure 9, the optimal extraction temperature of organic acids is between 120 and 180 °C, and the time is between 10 and 40 min.

### 4.6. Essential Oils

Essential oils, also called volatile oils or ethereal oils, refer to the volatile chemical compounds derived from plants. Essential oils often consist of the parts of the flowers, leaves, wood, bark, root, seeds, or peel. Ethereal oils are usually lipophilic and easily soluble in oils, which enables them to easily penetrate the skin and enter the body through the rich capillaries under the subcutaneous fat. Essential oils have been used in folk medicine in ancient China, Egypt, Arabia, and Greece throughout centuries. Therefore, some researches have attempted to utilize SBWE as a green solvent to extract essential oils from plants, and recent researches are illustrated in Table 6 [12,22,28,174,175,176,177,178,179,180,181,182,183,184,185,186,187,188,189,190,191,192,193,194,195,196,197,198,199,200,201,202,203,204,205,206]. The most common essential oils are extracted by distillation [174,176,180,181], *n*-hexane [174] or supercritical carbon dioxide [181], or dichloromethane [181].

Coriander (*Coriandrum sativum* L.) seeds contain an essential oil (up to 1%) and are increasingly used as condiment in the food industry. Eikani et al. [174] and Zeković et al. [175,181] have extracted and isolated the essential oil by SBWE. Extraction temperatures (100, 125, 150, and 175 °C), mean particle sizes (0.25, 0.50, and 1 mm), and water flow rates (1, 2, and 4 mL/min) were investigated by Eikani et al. Separation and identification of the components were carried out by GC-FID and GC-MS. They concluded that hydro distillation and Soxhlet extraction showed higher extraction efficiency, but the SBWE resulted in the essential oils more being concentrated in valuable oxygenated components. Zoran et al. also concluded that the SBWE process would have advantage in terms of time consumption since 20 min for this process was significantly lower than the 2 h, 4–5 h, and 4 h required for herbal decoction, Soxhlet, and supercritical fluid extraction, respectively.

Most essential oils extracted by SBWE have no research on the activities, except for Ma et al. [176], who extracted essential oils from *K. galangal* using ultrasound-enhanced SBWE and investigated the antioxidant ability of the essential oils. The effects of temperature, extraction time, pressure, ultrasonic power density, and frequency on the extraction yield were investigated. The antioxidant activity of the essential oils was evaluated by the assays of the 2,2-diphenyl-1-picryl-hydrazyl (DPPH) scavenging ability and the superoxide anion radical scavenging activity. The result demonstrated that antioxidant effects of UAE extracted essential oil were better than that of herbal decoction and SBWE. As shown in Figure 9, the optimal extraction temperature of essential oils is between 120 and 160 °C, and the time is between 10 and 35 min.

### 4.7. Alkaloids

Alkaloids are a class of basic nitrogen-containing organic compounds with a great structure diversity, most of which are heterocyclic compounds, and the nitrogen atom is in the heterocyclic ring. Most alkaloids are alkaline and have therapeutic and recreational activities. Alkaloid-containing plants have been used in folk medicine for centuries. Therefore, many researches are paid attention to the extraction and separation of alkaloids from plants. Traditional extractions include organic solvents, such as methanol [183,184], ethanol [183], or an acidic solution [187]. Recently, SBWE was successfully applied to the extraction or separation of alkaloids from plants or animals, as illustrated in Table 6.

Due to the diversity and relatively poor thermal stability of alkaloids, the optimal extraction methods depend on the physio-chemical characteristics of alkaloids. Liu et al. [16] used SBWE and capillary electrophoresis (CE) to extract and determine cytisine, sophocarpine, matrine, sophoridine, and oxymatrine in *Sophora flavescens* Ait., which is a traditional Chinese medicine. The extraction yields obtained using SBWE, ASE, UAE, and chloroform soaking extraction methods were compared. SBWE needs a short extraction time, as there is no need for organic solvent consumption and it exhibited the highest extraction efficiency for the total alkaloid yield. Similarly, Torto et al. [184] concluded that both SBWE and conventional methods achieved comparable extraction yields, while reflux and UAE were slower (over 6 h) and employed large quantities of organic solvents. Therefore, the SBWE method was simple and relatively fast for extraction. However, Liu et al. [183] showed that LC-ESI-obitrap MS provides a powerful method for the identification and determination of hepatotoxic pyrrolizidine alkaloids, and reflux showed a higher extraction efficiency compared with SBWE. Komes and his collaborators [186] investigated conventional and innovative extraction techniques (SBWE, UAE). They found both extracts of banana and beetroot peels obtained by THD (100 °C, 20 min) exhibited the highest total phenolic content and antioxidant capacity. Extraction by infusion (80 °C, 30 min) yielded a beetroot peel extract with the highest total betacyanin content. The optimal extraction temperature and time for alkaloids is also listed in Figure 9.

### 4.8. Quinones

The quinones are a kind of organic components that have quinone structures, which can mainly be divided into four types: benzoquinone, naphthoquinone, phenanthraquinone, and anthraquinone. Anthraquinone and its derivatives widely exist in plants and can be obtained from many plants, especially conifers. Some quinones have desirable pharmacological properties, such as purgative, antimicrobial and antiparasitic, anti-cardiovascular roles, etc. Extraction of quinones from natural products has utilized ethyl acetate [190], SFE [190], ethanol [191,193], or water as extraction solvents, shown in Table 6.

Machmudah et al. [188] performed extraction of phenolic compounds from pericarps of mangosteen by subcritical water treatment at temperatures and pressures of 120 to 160 °C and 1 to 10 MPa in batch and semi-batch systems. They added 10 to 30% deep eutectic solvent (DES) to subcritical water, and the results showed that with 30% DES, the yields of xanthone and phenolic compounds content were 24.87 mg/g dried sample and 179.54 mg of gallic acid equivalent/g dried sample at extraction temperatures of 160 and 120 °C in the batch system, respectively. The addition of DES in SBWE process could accelerate hydrolysis reaction to extract plant biomass components matrix.

*Morinda citrifolia* (Noni), planted in tropical Asia, has been used in folk remedies to treat various kinds of diseases and symptoms. Shotipruk and coworkers have conducted a series of experiments on this plant [191,192,193]. They used a continuous flow system to extract damnacanthal, alizarin, and 1,2-dihydroxyanthraquinone. They found that pressure had no significant effect on the results for the range 110–220 °C. Compared with conventional extraction, SBWE and Soxhlet resulted in extracts that have the highest antioxidant activity. The data were fitted with mathematic models to determine the extraction mechanism. The results suggested that the overall extraction mechanism was influenced by solute partitioning equilibrium with external mass transfer through liquid film.

The optimal extraction temperature and time for quinones in SBWE are shown in Figure 10. The optimal extraction temperature of quinones is between 160 to 170 °C and the time is between about 60 to 120 min.

### 4.9. Terpenes

Terpenes are a large and diverse class of organic compounds using isoprene as the basic structure unit, which widely exists in plants and some insects, and can be obtained from many plants, especially conifers. Terpenes have wide varieties and may be classified by the number of isoprene units in the molecule, such as monoterpenes, hemiterpenes, sesquiterpenes, diterpenes, triterpenes, tetraterpenes, and polyterpenes. Terpenes are of importance for the use in food, cosmetics, and pharmaceutical industries. However, the extraction of terpenes and terpenoids from natural products is often problematic, illustrated in Table 6. Consequently, they are mainly produced by industrial synthesis, usually from petrochemicals.

Chen et al. [197] employed SBWE for the efficient extraction sesquiterpene lactones from I. racemose. Extraction time (23.2–56.8 min), temperature (129.5–230.5 °C) and flow rate (1.3–4.7 mL/min) were investigated. A comparison of SBWE with traditional extraction technologies (Soxhlet, UAE, and SFE) showed that subcritical water could be a green and efficient substitution for the extraction of sesquiterpene lactones from I. racemose. Xiao et al. [13] developed SBWE for extraction of ursolic acid from *Hedyotis diffusa*. The RSM model proved to predict the experimental results very well and demonstrated that UA yield was mainly dependent on solvent/solid ratio, followed by particle size and temperature. Four extraction methods (UAE and SBWE) were comparatively analyzed, which indicated that SBWE was a time-saving, cost-saving, and environment-friendly extraction technology for the extraction of UA from *Hedyotis fusa*. Other researchers achieved the same experimental results, except for Falev et al. [200], who found that subcritical water is a poor solvent for pentacyclic triterpenes. Extraction with subcritical solvents (aliphatic alcohols, acetonitrile, and chloroform) is the most rapid and efficient way to isolate pentacyclic triterpenes from plant raw materials.

The optimal extraction temperature and time for terpenes in SBWE are summarized in Figure 10. The optimal extraction temperature of quinones is between 130 and 225 °C, and the time is between about 10 and 50 min.

### 4.10. Lignans

Lignans are a kind of natural compound formed by the polymerization of two molecules of phenylpropanine derivatives (i.e., C_3_-C_6_ monomers), which exist in plants and belong to phytoestrogens. The monomers composing of lignans include cinnamic acid, cinnamyl alcohol, acryl benzene, allylbenzene, and so on. Lignans are reported to have potential antitumor, anti-inflammatory, or antioxidant activity in the laboratory models of human diseases. Most lignans are lipophilic, and easily soluble in organic solvents. Therefore, extraction of lignans from plants often utilized ethanol, ether, or acetone as extraction solvents. Researches about SBWE of lignans are only a few papers, as listed in Table 6.

Evrim [48,50] has conducted a detailed investigation of the material shape (flaxseed, ground flaxseed meal, and flaxseed meal sticks), temperature, extraction time, pressure, fresh water, and sample amount on the effect of secoisolariciresinol diglucoside lignan content using accelerated solvent extractor. The highest amount (12.94 mg/g) and extraction yield (72.57%) were obtained at 180 °C for 15 min, 10.3 MPa, and 40% fresh water using 5 g of flaxseed meal sticks. Bodoira et al. [42] used water and ethanol under sub-critical conditions to extracted bioactive compounds from sesame (*Sesamum indicum* L.) defatted seeds. At 220 °C, 8 MPa and 63.5% ethanol as co-solvent condition, the yields of lignans, TP, flavonoids compounds were maximized, and the antioxidants were similar to those reached by using synthetic antioxidants. Kinetic studies showed a high extraction rate of phenolic compounds until the first 50 min of extraction, and it was in parallel with the highest scavenging capacity. SBWE could selectively extract different kinds of bioactive compounds only by changing process conditions. The optimal extraction temperature for lignans is about 180 °C, as shown in Figure 10.

### 4.11. Steroids

A steroid is a class of natural chemical components widely existing in nature, including phytosterin, bile acids, C_21_ steroids, insect allergic hormones, cardiac glycosides, steroidal saponins, steroidal alkaloids, bufogenin, etc. The steroid core structure has the basic skeleton structure of cyclopentano-perhydrophenthrene, bonded in four “fused” rings: three six-member cyclohexane rings and one five-member cyclopentane ring. Steroid compounds have various biological activities. They have a wide range of applications. The extraction methods of steroids from natural products include maceration, Soxhlet, and SBWE are illustrated in Table 6.

Ginseng is a well-known traditional Chinese medicine with numerous pharmacological effects [24,25,26]. These bioactive components are mainly the ginsenosides, polyphenols, amino acids, and polysaccharides. Renata et al. [24] utilized SBWE to obtain fructooligosaccharides and beta-ecdysone from Brazilian ginseng root and aerial parts. Lee et al. [25] extracted red ginseng by varying the temperature (150–200 °C) and extraction times (5–30 min) in SBWE. Compared to traditional heat extraction methods (ethanol, hot water, and methanol), extracts of red ginseng from SBWE had higher ginsenoside concentrations and antioxidative properties. Shivraj et al. [204] applied subcritical water to extract withanosides and withanolides from ashwagandha at varying temperatures (100–200 °C) and extraction times (10–30 min). Various biological effects, including cytotoxicity, antioxidant, and enzyme inhibitory activities were quantified using HPLC. Withaferin A showed significant reduction in cell viability of cervical cancer cells, with IC_50_ values 10 mg/mL and 8.5 μM/mL, respectively, but no cytotoxic effect for normal cells. Thus, SBWE could be used for extraction of pharmacologically active fractions with therapeutic applications. The optimal extraction temperature for steroids is about 160 °C, as shown in Figure 10.

## 5. Conclusions and Future Perspectives

Due to its nontoxic, nonflammable, and widely available nature, SBWE of natural products has gained greater attention during the last decade. Our review of over 200 articles shows that SBWE is a promising technology in extracting natural products. A wide variety of plant-related materials have been extracted by subcritical water. Sample materials include medicinal herbs, seasoning herbs, vegetables, fruits, algae, shrubs, tea leaves, grains, and seeds. The following natural products have been extracted by SBWE: Alkaloids, carbohydrates, essential oil, flavonoids, glycosides, lignans, organic acids, polyphenolics, quinones, steroids, and terpenes. Both static and dynamic mode are employed in SBWE. In general, the static extraction efficiency is lower than that of the dynamic mode. Thus, most SBWE experiments were conducted using the static mode, and then, followed by the dynamic mode. SBWE system was also coupled with HPLC, subcritical water chromatography, and high-temperature liquid chromatography. Co-solvents such as ethanol, methanol, salts, and ionic liquids were also used to improve SBWE efficiency. It should be noted that temperature has the most significant effect on SBWE efficiency, and thus, it can be optimized. The optimal temperature ranges from 120 to 200 °C for extracting the natural products mentioned above.

A major advantage of SBWE of natural products is that water is nontoxic, and therefore it is more suitable for the extraction of herbs, vegetables, and fruits since the extracts can be safely consumed by human or animals. In addition, if no organic modifiers are used in SBWE, the liquid waste generated after SBWE does not require waste disposal. However, the high temperature used in SBWE may potentially cause analyte degradation. Thus, one must evaluate analyte stability under the temperature conditions to be used in SBWE to ensure that the analytes extracted do not undergo degradation during SBWE. SBWE employs high-pressure and high-temperature fluid, and great caution is required to ensure safe operation of the SBWE system. In addition, frequent pluming blockage may occur during dynamic SBWE process.

A vast majority of SBWE research reported is studied at the bench scale. The next level of SBWE should be scaling up to the industrial level. Results of some pilot scale studies have demonstrated the potential development of large scale SBWE processes.

## Figures and Tables

**Figure 1 molecules-26-04004-f001:**
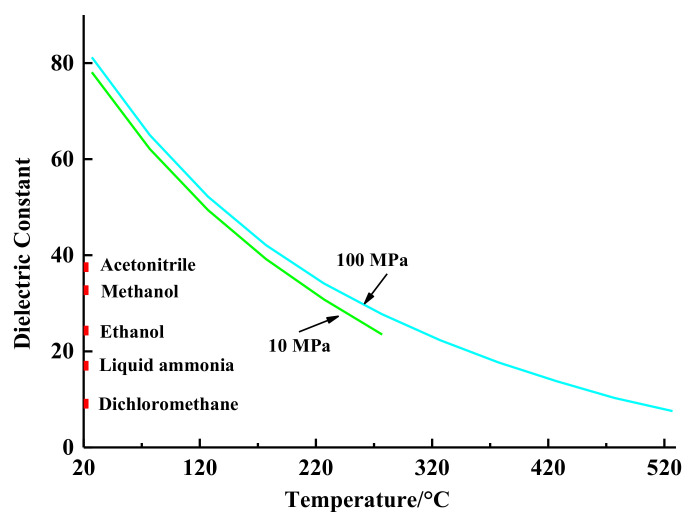
Dielectric constant of water at 27 to 527 °C and 10 to 100 MPa, acetonitrile, methanol, ethanol, liquid ammonia, and dichloromethane at 20 °C and 0.1 MPa.

**Figure 2 molecules-26-04004-f002:**
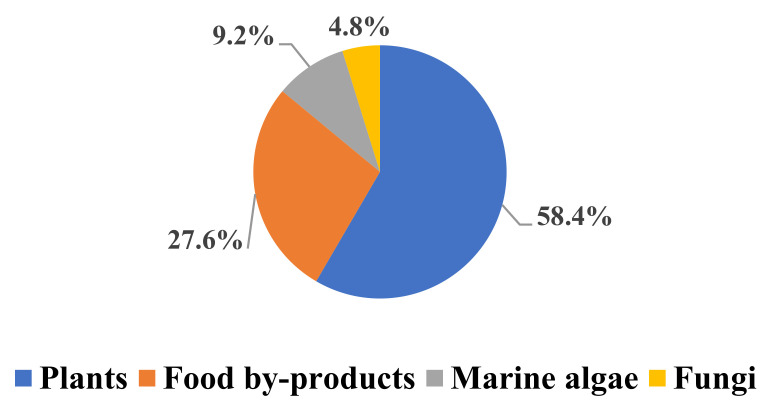
Types of sample matrices extracted by subcritical water.

**Figure 3 molecules-26-04004-f003:**
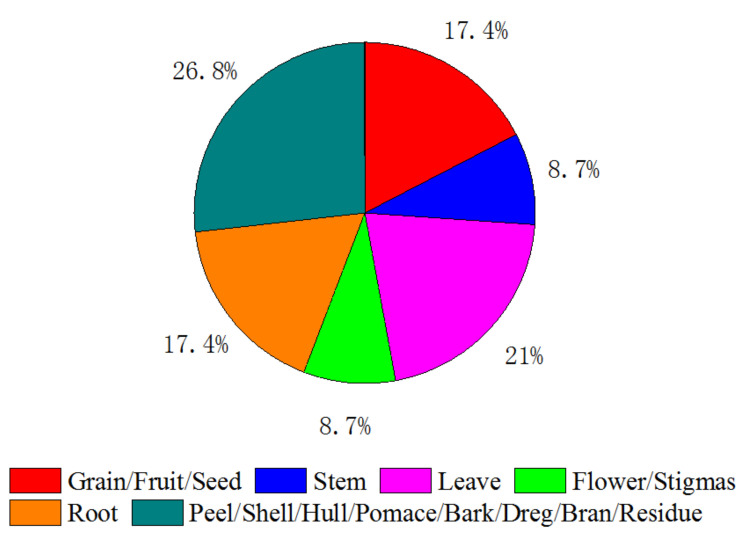
Parts of medicinal herbs extracted by subcritical water.

**Figure 4 molecules-26-04004-f004:**
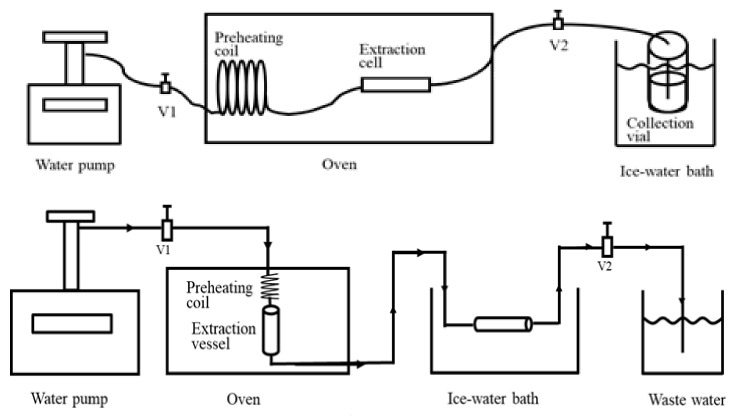
Subcritical water extraction system without solid trapping (top) and with solid trapping (bottom).

**Figure 5 molecules-26-04004-f005:**
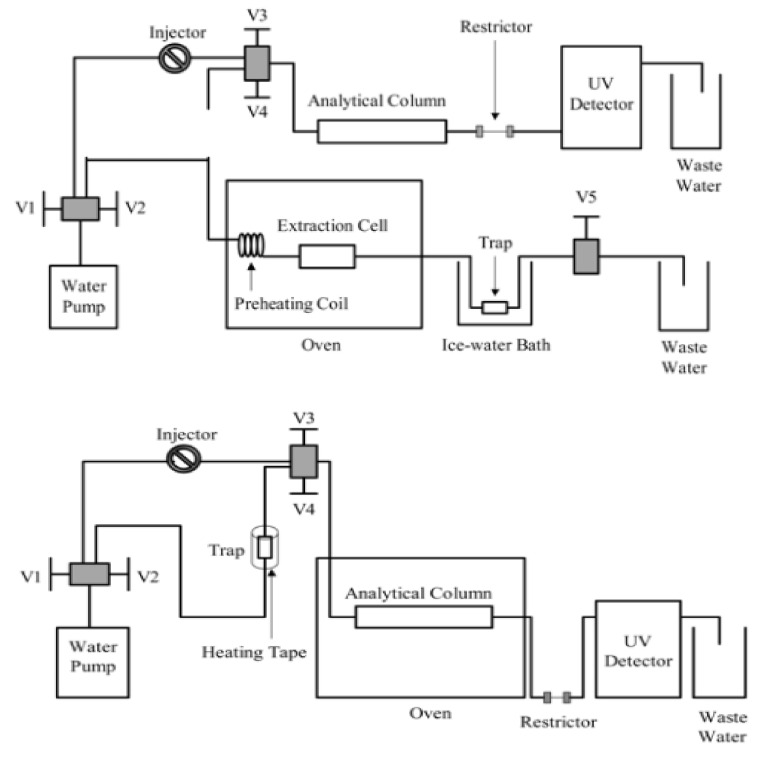
Offline coupling of SBWE with SBWC. Adapted with permission from [70] (Lamm, L.; Yang, Y. Off-line coupling of SBWE with subcritical water chromatography via a sorbent trap and thermal desorption. *Anal. Chem*. **2003**, *75*, 2237–2242.). Copyright 2003 American Chemical Society.

**Figure 6 molecules-26-04004-f006:**
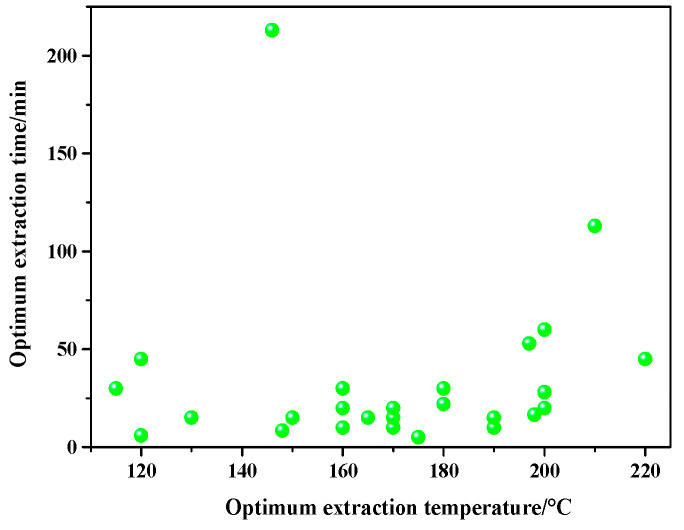
Optimum extraction conditions flavonoids.

**Figure 7 molecules-26-04004-f007:**
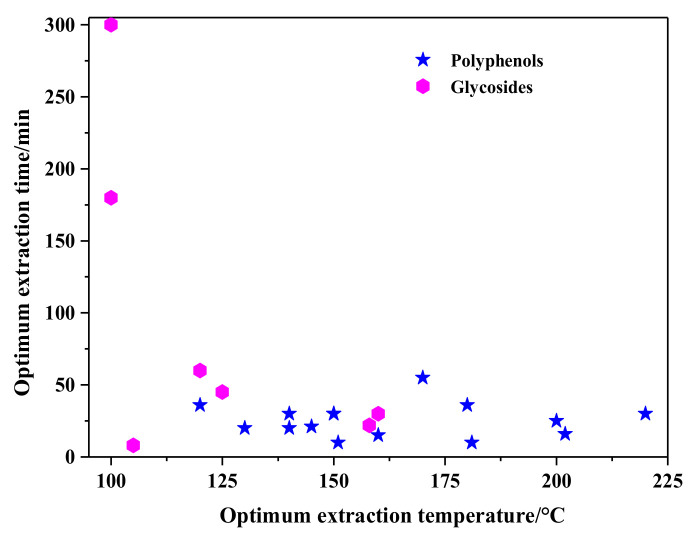
Optimum extraction conditions for polyphenols and glycosides.

**Figure 8 molecules-26-04004-f008:**
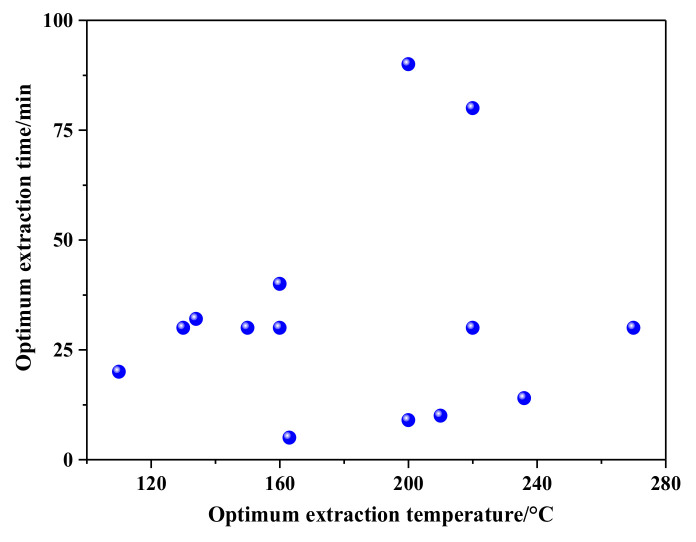
Optimum extraction conditions for organic acids.

**Figure 9 molecules-26-04004-f009:**
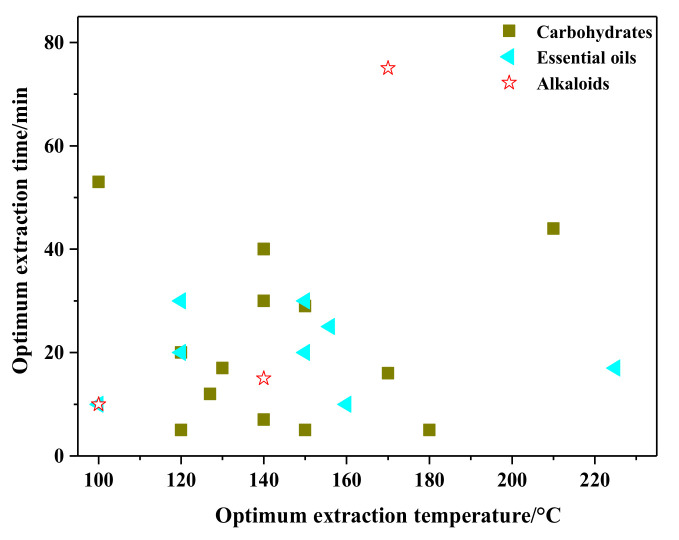
Optimum extraction conditions for carbohydrates, essential oils, and alkaloids.

**Figure 10 molecules-26-04004-f010:**
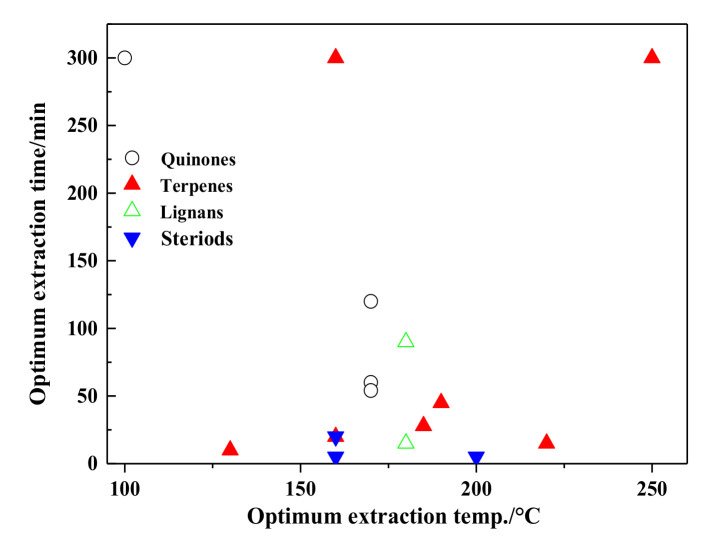
Optimum extraction conditions for quinones, terpenes, lignans, and steroids.

**Table 1 molecules-26-04004-t001:** SBWE of flavonoids.

Samples	Medicinal Parts	Compounds Extracted	Extracts Activity	Extraction Conditions	Analytical Methods	Other Extraction Methods (Solvent, Ratios of Yields)	Ref.
*Panax ginseng* C.A. Meyer	stem leave	TP and flavonoids	antibacterial	110 and 165 °C, 15 min 190 °C, 10 min	TEM, UV	heating (water 95.4%; ethanol 91.3%)	[15]
*Chamomilla matricaria* L.	flowers	TP, TF, 18 polyphenolic compounds, apigenin	antioxidant, enzyme inhibitory activity	65–210 °C, 5–60 min 1:30–1:100 g/mL	TLC, UV, HPLC-MS		[18]
*Allium cepa*	onion wastes	quercetin-4′-glycoside, quercetin, etc.		40–160 °C, 5 min, 5 MPa, 1–10 mm, pH 3.0–7.0	LC-MS/MS HPLC-UV	convention (methanol and hydrochloric acid 94.3%)	[19]
*Crocus sativus* L.	stigmas	TP, dodecane, γ-terpinene, tetradecane, etc.	antioxidant (DPPH, FRAP), antibacterial	100–180 °C, 10–30 min, 1:10 g/mL	GC/MS, UV-vis		[21]
*Saururus chinensis*, etc.	skin, leave, peel, etc.	quercetin, isorhamnetin, kaempferol, isoquercitrin, etc.		10 MPa, 110–200 °C, 5–15 min	HPLC		[30]
*Camellia sinensis*	leaves	epigallocatechin gallate		80–120 °C, 3–7 min, 40–60 mL/g	HPLC	convention (water 87.6%)	[44]
*Origanum vulgare* L.	leaves	TP, flavanone, flavone, flavanol	antioxidant (DPPH, TEAC, ABTS)	10.34 MPa, 30 or 15 min 25–200 °C	HPLC- DAD, UV		[47]
orange	peels	reducing sugar, TP, pectin, hesperidin, narirutin	antioxidant (DPPH, FRAP)	110–150 °C, 10–30 mL/min 10 MPa	HPLC, UV-vis	Soxhlet (ethanol 79.2%), shaker (ethanol 250%), UAE (ethanol 114%)	[50]
orange	peels	flavones, 7-hydroxyflavone		100–150 °C, 0.5 mL/min	GC-FID	UAE (methylene chloride)	[70]
*Citrus unshiu* Markovich	peels	rutin, naringin, hesperidin, naringenin		0.5–14 MPa, 5–15 min, 100–190 °C	HPLC		[74]
*Allium cepa* L.	peels	TP, TF, quercetin	antioxidant (DPPH, TBA, FTC)	110 and 165 °C, 15 min, p < 3.4 MPa	HPLC, UV	heating (ethanol 153%; water 45.6%)	[75]
*Hippophae rhamnoides*	leaves	TP, TF, isorhamnetin, kaempferol, quercetin	antioxidant, cytotoxicity	25–200 °C, 15 min, 10.34 MPa	HPLC, UV, FM	maceration (water 21.3%), Soxhlet (ethanol 64.6%)	[76]
*Allium cepa* L.	peels	TP, TF, kaempferol, quercetin	antioxidant (DPPH)	170–230 °C, 3 MPa, 30 min, pH 2–10	HPLC, UV-vis	heating (ethanol 26.7%)	[77]
*Achillea millefolium* L.	herbal dust	TP, TF, HMF, chlorogenic acid	antioxidant (DPPH, TEAC, ABTS)	120–200 °C, 10–30 min 0–1.5% HCl, 3 MPa	HPLC, UV-vis		[78]
*Curculigo latifolia*	root	TP, TF, pomiferin, etc.	antioxidant (DPPH, ABTS, TEAC)	100–200 °C, 10 MPa 30–120 min, 0.5 mL/min	LC-MS, UV		[79]
*Citrus unshiu*	peels	hesperidin and narirutin		110–190 °C 3–15 min	HPLC		[80]
*Glycine* max	okara	genistin, daidzin, genistein, daidzein		100–200 °C, 5 min, 2–5 MPa, 10–30 g/mL	HPLC	Soxhlet (methanol, 108%)	[81]
onion	skins	quercetin, quercetin-4′-glucoside		100–190°C, 5–30 min, 9–13 MPa	HPLC	convention (methanol, 92.8%)	[82]
*Puerariae lobata*	root	puerarin, daidzin, daidzein 3-methoxypuerarin		100–200 °C, 15–75 min 1:10–1:25 g/mL	HPLC	reflux (ethanol 91.6%), UAE (water 95.9%)	[83]
*Coriandrum sativum*	seeds	TP, TF	antioxidant (DPPH)	100–200 °C, 10–30 min 3–9 MPa	UV		[84]
*Citrus unshiu*	peels	flavanones, polymethoxy-Flavones, etc.	anticancer, cardioprotectives	120–180 °C, 1.0–2.0 mL/min, 5.0 ± 0.1 MPa	GC, HPLC,	convention (methanol 75.0%; ethanol 41.6%; acetone 17.2%)	[85]
*Phlomis umbrosa*	whole part	TP, TF, iridoids glycosides	antioxidant (DPPH, ABTS)	110–200 °C, 10 MPa, 1–25 min	HPLC, ESI-MS	convention (ethanol; methanol; water)	[86]
*Actinidia deliciosa*	peels	TP, TF,	antioxidant (DPPH, ABTS, FRAP)	120–160 °C, 0–30 min, 3 MPa, pH 2–5.5	UV-vis, pH	convention (ethanol 81.9%)	[87]
*Scutellaria baicalensis*	root	baicalin, baicalein, wogonin, wogonoside		110–160 °C, 10–90 min, 20–100 mesh	HPLC	HRE (ethanol 93.0%)	[88]
*Citrus unshiu*	pomaces	TP, polymethoxylated flavones, sinensetin, etc.	antioxidant (DPPH, TP)	25–250 °C, 10–60 min, 0.1–5.0 MPa	HPLC, UV		[89]
*citrus unshiu*	peels	hesperidin, narirutin, prunin, naringenin, sinensetin, etc.	antioxidants (DPPH, FRAP), enzyme	145–175 °C, 15 min 5 MPa, 0.75–2.2 mL/min	HPLC	2M HCl extraction 42.9%; 2 M NaOH extraction 38.9%	[90]
*citrus unshiu*	peels	hesperidin and narirutin		110–200 °C, 5–20 min, 10 ± 1 MPa	HPLC, MS/MS	convention (ethanol 56.4%; methanol 35.8%; water 6.2%)	[91]
*palatiferum* Radlk.	leaves	TP, TF, protein, saponin, sugar, apigenin, kaempferol	antioxidants (DPPH, FRAP, ABTS),	110–270 °C, 15 min, 8 MPa 1:70 g/mL	HPLC, UV	convention (water 77.7%; methanol 32.8%), Soxhlet (ethanol 43.7%)	[92]
*Glycyrrhiza uralensis* Fisch.	root	TP, TF, liquiritin, flavanone, isoflavone	antioxidants (DPPH, ABTS)	80–320°C, 2–100 min, 7.0 MPa, 1:30 g/mL, pH 3–11	HPLC, MS/MS, UPLC	UAE (water 20.6%; ethanol 44.9%), MAE (water 25.6%; ethanol 63.8%)	[93]
*Tagetes erecta* L.	flower residues	TP, TF, 5-HMF, reducing sugar, free amino acids	antioxidants (DPPH, ABTS)	80–260 °C, 15–90 min 1:20–1:60 g/mL,120 rpm	HPLC-DAD, UV	leaching (water 9.4%; methanol 69.9%; ethanol 68.8%; acetone 94.0%), UAE (water 9.9%; methanol 69.8%; ethanol 64.3%; acetone 87.6%)	[94]
*Daucus carota*	leaves	polyphenols, luteolin		110–230 °C, 0–114 min, 4 MPa	UV, PLC		[95]
*Matricaria chamomilla* L.	flowers	TP, TF, apigenin-7-O-glucoside, etc.	antimicrobial, cytotoxic activity	200 °C, 40 min, 1:50 g/mL	UHPLC, HESI- MS/MS, UV	Soxhlet (ethanol 129%), MAE (ethanol 117%), UAE (ethanol 104%)	[96]
*Silybum murianum* L	seeds	taxifolin, silychristin, silydianin, and silybin		75–250 °C, 40–60 min, 12.5 MPa, 0.1–0.5 mm	HPLC	convention (ethanol 101%; water 43.6%)	[97]
*Echinacea purpurea* L.	flowers	TP, TF	antioxidant (TEAC, ABTS)	103.4–216.56 °C, 3 MPa, 5.86–34.414 min	UV-vis		[98]
*Humulus lupulus*	pellets	TP, desmetylxanthohumol, prenylflavonoids, etc.	anti-inflammatory	50–200 °C, 30 min, 10 MPa	HPLC, MS/MS	convention (hexane 17.2%; ethanol 105%)	[99]
*Kunzea ericoides*	leaves	TP, TF, 5-HMF, quercetin, catechin, syringic acid, etc.	antioxidant (DPPH, FRAP)	150–210 °C, 0–40 min 15–35 g/mL, 4 MPa	HPLC, UV	convention (ethanol 37.5%)	[100]
*Pistacia atlantica* subsp. *mutica*	hull	TP, kaffesaure, ethyl vanillin, flavanomarein, etc.	antioxidant (DPPH), reducing power	110–200 °C, 30–60 min, 10–50 g/mL	HPLC-DAD, UV	HWE (85 °C 42.8%)	[101]
*Satureja hortensis* L.	whole part	TP, TF, rosmarinic acid, rutin, quercetin, etc.	cytotoxic, antibacterial	140 °C, 30 min 4 MPa, 1:20 g/mL	HPLC-PDA, UV	maceration (ethanol 57.2%), Soxhlet (ethanol 18.4%), UAE (ethanol 69.2%), MAE (ethanol 81.3%)	[102]
*Urtica dioica* L.	leaves	TP, TF, twenty-seven compounds	cytotoxic, antifungal, antimicrobial	125 °C, 30 min, 3.5 MPa, 1:30 g/mL	UHPLC-HESI-MS/MS	UAE (water 48.5%), MAE (water 100%)	[103]
*Chamomilla recutita* R.	flowers	2 flavonoids, 4 esters, 1 amino acid, 11 phenols, etc.		150 or 200 °C, 5.0 ± 0.1 MPa, 1.7 mL/min, 40 min	UV, HPLC, GC-MS		[104]
*Glycine* max	okara	TP, gallic acid, syringic acid, ferruric acid, etc.	antioxidant (ABTS, DPPH, FRAP)	150 °C, 4 MPa, 5–275 min 20 mg/mL	UV, HPLC		[105]
*Carménère* grape	pomace	flavanols, stilbenes, and phenolic acids		90–150 °C, 5 min, 10 MPa, 15–50% glycerol	UPLC-MS		[106]
*Zingiber officinale*	root	TP, TF, four macro- and five microelements	antioxidant (OH·, ABTS, TRP, etc.)	80–180 °C, 1 h, 5MPa, 1:10 g/mL	UV-vis, ICP-MS	convention (water, 62.5%)	[107]
*Momordica foetida*	leaves	quercetin, kaempferol, isorhamnetin		100–300 °C, 5 mL/s 6.9± 1.4 MPa psi	UHPLC-q-TOF-MS		[108]

**Table 2 molecules-26-04004-t002:** Subcritical water extraction of polyphenols.

Samples	Medicinal Parts	Compounds Extracted	Extracts Activity	Extraction Conditions	Analytical Methods	Other Extraction Methods (Solvent, Ratios of Yields)	Ref.
*Allium ursinum* L.	leaves	TP, TF, 5-HMF, catechin, p-cumaric, ferulic acids, etc.	antioxidant (DPPH, ABTS), Millard products	120–200 °C, 10–30 min, 0–1.5% HCl, 1:10 g/mL	HPLC-DAD		[31]
*Terminalia chebula*	fruits	TP, allic acid, corilagin ellagic acid	antioxidant (ABTS)	120–220 °C, 2–4 mL/min, 4 MPa	TLC, UV, MS, NMR, HPLC	Soxhlet (water 74.5%; ethanol 46.3%), HWE (water 46.3%)	[34]
*Lycium ruthenicum* Murr.	fruits	total anthocyanin, seven anthocyanins	antioxidant (ABTS, DPPH)	110–170 °C, 30–90 min, 1–3 min/L	HPLC, UPLC-MS	UAE (water 59.8%; methanol 81.1%)	[35]
*Punica granatum* L.	peels	TP, TF, punicalin, etc.		100–220 °C, 5–30 min, 3.0 MPa	UV-vis, HPLC	MAE (water 121%; ethanol 146%)	[109]
*Castanea sativa*	shells	tannins, phenolic acids, flavonoids, anthocyanins	antioxidant (DPPH, FRAP, ABTS)	51–249 °C, 6–30 min	UV-vis, LC/ESI-MS		[110]
*Salvia officinalis* L.	by–products	TP, TF	antioxidant (DPPH, TEAC, reducing power)	120–220°C, 10–30 min, 3 MPa, 0–1.5% HCl	UV	maceration (water 59.9%)	[111]
*Pistacia vera* L.	hulls	gallotannin, anacardic acid, etc.	antioxidant (ABTS, FRAP)	110–190 °C, 6.9 MPa, 4 mL/min	HPLC-ESI/MS*^n^*	UAE (methanol 83.9%))	[112]
*Zingiber officinale*	pulp and peel	6-gingerol, 6-shogaol	antioxidant (FRAP)	10 MPa, 110–190 °C, 5–40 min	HPLC	convention (methanol 114%; water 77.1%)	[113]
*Sorfhum bicolor* L.	bran	TP, oligomeric procyanidins, taxifolin, taxifolin hexoside	antioxidant (DPPH, ABTS), antiproliferative	110–190 °C, 5–40 min, 1:10–1:50 g/mL	HPLC, ESI-MS/MS	heating (water 74.9%)	[114]
*Nelumbo nucifera*	seed epicarp	TP, proanthocyanidin dimers, trimer, cyanidin, etc.	antiproliferation effect (MTT)	100–180 °C, 5–25 min, 1:20–1:60 g/mL, 1–5‰ NaHSO_3_	HPLC-ESI-MS, UV	HWE (water 33.9%)	[115]
German chamomile	flowers	9 phenolic acids and derivatives	antioxidant, cytotoxic, enzyme	100 °C, 1–9 MPa, 30 min	UHPLC-DAD, MS/MS		[116]
*Fagopyrum tataricum*	grains	phenols, 13 phenolics, 4 flavonoids, 3 anthocyanins	antioxidant (TEAC, CAA and FRAP), cytotoxicity	220 °C, 60 min, 5 MPa, 1:60 g/mL	HPLC-MS, UV	UAE (water 83.5%)	[117]
*A. uva–ursi*	herbal dust	TP, TF	antioxidant (DPPH, reducing power)	120–220 °C,3 MPa, 10–30 min, 0–1.5% HCl	UV	maceration (water 38.5%; ethanol 69.5%)	[118]
*Hippophaë rhamnoides* L.	seed residue	TP, TF, proanthocyanidins	antioxidant (DPPH)	80–180 °C, 15–90 min, 1:10–1:50 g/mL, 6 MPa	UV	convention (water 19.6%; methanol 104%; ethanol 80.0%)	[119]
grape (Croatina)	pomace	TP, TF	antioxidant (DPPH)	100–140 °C, 8–15 MPa, 1–2 mL/min	UV	convention (water 5.3%; ethanol 7.87%)	[120]
*Matricaria chamomilla* L.	flowers	polyphenolic compounds, etc.	antioxidant, cytotoxic, enzyme inhibitory	65–210 °C, 30 min, 4.5 MPa	UHPLC-ESI-MS/MS, UV		[121]
*Nelumbo nucifera*	seedpods	TP, TF, proanthocyanidin dimer, isoquercetin, etc.	antioxidant, antiproliferative (HepG2)	100–180 °C, 30–70 mL/g, 5–25 min, 1–6‰ NaHSO_3_	UV-Vis, HPLC, ESI-MS^n^	HWE (water 91.4%)	[122]
*Vitis vinifera* L.	grape pomace	catechins, flavonols, tannins, proanthocyanidins, etc.	antioxidant (DPPH, ABTS)	40–120 °C, 10 min, 10.34 MPa, 10–40% NADES	UV, HPLC-ESI-MS		[123]
sweet chestnut	bark	TP, tannins, ellagic and gallic acids, ellagitannins, etc.	antioxidant (DPPH)	150–250 °C, 10–60 min, 10–30 mL/g, 4.5 MPa	UV-Vis, HPLC		[124]
*Symphytum officinale*	root	TP, TF	antioxidant (DPPH), enzyme inhibitory	120–200 °C,10–30 min, 0–1.5% HCl	UV, ELISA	UAE (methanol 2.5%; ethanol 17.4%); maceration (methanol 4.4%; ethanol 29.8%)	[125]
*Pinot Nero*	grape skins	TP		80–120 °C, 2 h,10 MPa, 2–5 mL/min	UV-Vis		[126]
*Coffea arabica* L.	spent coffee grounds	TP, caffeoylquinic acid, feruloylquinic acid, etc.	antioxidant (DPPH, ABTS)	160–180 °C, 35–55 min, 14.1–26.3 g/L	HPLC-ABTS^+^, MS, UV		[127]
*Curcuma**longa* L.	rhizomes	curcumin, demethoxycurcumin		120–160 °C, 6–22 min, 1–2.5 MPa	HPLC-UV, SEM		[128]
*Curcuma longa* L.	rhizomes	α-phellandrene, curcumin, β-caryophyllene, trans-β-farnesene, β-bisabolene, γ-curcumin, etc.		90–150 °C, 1–4 mL/min, 2 MPa, 0.5–1.5 mm	GC/GC-MS, GC -FID	HD (80.7%), Soxhlet (*n*-hexane 1.2-fold)	[129]
*Curcuma longa* L.	rhizomes	curcumin, demethoxycurcumin, bisdemethoxycurcumin		110–150 °C, 1–10 min, 0.5–10 MPa	HPLC	convention (ethanol, 1.13-fold)	[130]
*Curcuma longa* L.	rhizomes	curcumin, demethoxycurcumin, bisdemethoxycurcumin		90–250 °C, pH 1.0–5.5 5.0 MPa, 0.5 mL/min	HPLC, UPLC, LC-MS	Soxhlet (acetone, 1.17-fold)	[131]

**Table 3 molecules-26-04004-t003:** Subcritical water extraction of organic acids.

Samples	Medicinal Parts	Compounds Extracted	Extracts Activity	Extraction Conditions	Analytical Methods	Other Extraction Methods (Solvent, Ratios of Yields)	Ref.
*Panax ginseng* Meyer	root	TP, chlorogenic acid, caffeic acid, gallic acid, etc.	antioxidant (DPPH, ABTS, FRAP, HRS)	100–240 °C, 15 min, 4–9 MPa, 200 rpm	HPLC, UV		[23]
*Helicteres isora* L.		hexadecanoic acid, octadecnoic acid, heptadecen-8-carbonic acid etc.	antibiofilm, antioxidant, antimicrobial, antienzymatic	160 °C, 30 min, 1 MPa, 1: 30 g/mL	GC-MS, UV		[27]
XiLan olive fruit	olive dreg	TP, chlorogenic acid, gallic acid, syringic acid, etc.	antioxidant (ABTS, DPPH, reducing power)	100–180 °C, 5–60 min, 1:20–1:60 g/mL	LC-MS-IT-TOF, UV	convention (methanol 3.2%; ethanol 0.6%; DMK 0.9%)	[33]
*Camellia oleifera* Abel.	seeds	free fatty acids (palmitic acid, stearate, oleic acid, etc.), tea saponin	antioxidant (DPPH)	60–160 °C,2–7 MPa, 5–60 min, 1:3–1:25 g/mL	GC-MS, FT-IR	Soxhlet (petroleum ether 100%), cold pressed (100%)	[37]
sunflower seeds (Natura)	dehulled seeds	total proteins, total carbohydrates, TP	antioxidant capacities	60–160 °C, 5–120 min, 3 MPa, 1:10–1:30 g/mL	GC-FID, UV-Vis, HPLC	Soxhlet (hexane 67.3%)	[38]
cottonseed (Egypt)	cottonseed	linoleic acid, palmatic acid, oleic acid, myristic acid		180–280 °C, 5–60 min, 1:2–2:1 g/mL	GC-FID,	heating (hexane 89.5%)	[39]
green coffee (Robusta Uganda)	beans	chlorogenic acid		130–170 °C, 40–90 min, 0–30 % ethanol	HPLC	convention (ethanol 66.7%)	[45]
*Nannochloropsis gaditana*		fatty acids, omega-3, omega-6, lipid		156.1–273.9 °C, 6.6–23.4 min, 33–117 g/L	GC-FID, SEM	Soxhlet (n-hexane 100%)	[51]
*Saccharina japonica*		gallic, caffeic, vanillic, syringic, chlorogenic, p-hydroxybenzoic acids, etc.	antioxidant (DPPH, ABTS, total antioxidant (FRAP)	100–250 °C, 5 min, 5 MPa, 0.25–1.00 M ILs	HPLC, UV	convention (DMK 0.2%; DCM 0.3%; Et_2_O 0.8%; IL 1.6%)	[52]
*Haematococcus pluvialis*		p-hydroxybenzoic acid, gallic acid, siringic acid, vanillic acid, etc.	antioxidant (ABTS, TEAC), antimicrobial activity	50–200 °C, 20 min, 10 MPa	HPLC-DAD-MS, SEM, GC-MS		[53]
*Momordica charantia*	fruits	TP, gallic acid, gentisic acid, chlorogenic acid	antioxidant (ABTS)	130–200 °C, 10 MPa, 2–5 mL/min	HPLC, UV	Soxhlet (methanol 4.9%), UAE (methanol 4.0%)	[132]
*Morus nigra* L., *Teucrium chamaedrys* L., *Geranium macrorrhizum* L., *Symphytum officinale* L.	leaves, flowers	TP, chlorogenic acid, gallic acid, vanillic acid, etc.	antioxidant, antifungal, antibacterial, cytotoxic	60–200 °C, 30 min, 1 MPa, 1:40g/mL	HPLC-DADUV		[133]
*Prunus avium* L., *Prunus cerasus* L.	stems	3 alcohols, 10 organic acids, etc.	antioxidant, antiproliferative	150 °C, 30 min, 2 MPa	GC-MS, UV		[134]
*Castanea sativa*	nuts	ellagic acid, feru lic acid, gallic acid, etc.	antioxidant	120–135 °C, 15–60 min	HPLC		[135]
*Solanum tuberosum*	potato peel	TP, gallic acid, caffeic acid, chlorogenic acid, protocatechuic acid, etc.		100–240 °C, 30–120 min, 6 MPa	HPLC, UV	convention (methanol 1.6%; ethanol 2.0%)	[136]
*Actinidia deliciosa*	pomace	TP, chlorogenic acid, protocatechuic acid, etc.	antioxidant (DPPH, FRAP, ABTS)	170–225 °C 10–180 min, 5 MPa	UV, HPLC, pH		[137]
*hypnea musciformis*		chlorogenic, protocatechuic, and gallic acids, TP, TF, etc.	antioxidant (DPPH, ABTS), emulsify	120–270 °C, 10 min, 1:50–1:150 g/mL	pH, UV, HPLC		[138]
*Carica papaya* L.	papaya seeds	TP, 18 phenolic acids, 20 flavonoids, 1 stilbene, etc.	antioxidant (DPPH, β-carotene bleaching)	70–150 °C, 10 MPa, 1–40 min, 4 mL/min	LC-ESI-MS/MS, UV	Soxhlet (water 37.1%)	[139]
*Zingiber officinale*	ginger rhizome	12 sugars, 8 diols, 4 phenolic acids, etc.	antimicrobial, cytotoxic	150 °C, 1 h, 1:10 g/mL	HPLC-ESI-TOFMS	heating (water)	[140]
*Chlorella* sp. microalgae		TP, caffeic acid, ferulic acid, p-coumaric acid	antioxidant (DPPH)	100–250 °C, 5–20 min	UV, SEM, HPLC		[141]
*Vitis vinifera*	vine-canes	TP, flavonoids, phenolic acids, flavonols	antioxidant, antiradical	125–250 °C, 50 min	HPLC, UV		[142]
*Cinnamomum Cassia Blume*	cinnamon	coumarin, cinnamic acid, cinnamaldehyde, cinnamyl alcohol, etc.		110–130°C, 20–60 min, 2–4 MPa, 1:10 g/mL	HPLC		[143]

**Table 4 molecules-26-04004-t004:** Subcritical water extraction of glycosides.

Samples	Medicinal Parts	Compounds Extracted	Extracts Activity	Extraction Conditions	Analytical Methods	Other Extraction Methods (Solvent, Ratios of Yields)	Ref.
*Phaleria macrocarpa*	fruits	mangiferin		323–423 K, 1–7 h, 0.7–4.0 MPa	HPLC, LC-MS	convention (water 69.6%; ethanol 34.1%; methanol 108%), HRE (water 85.7%; ethanol 60.8%; methanol 115%), Soxhlet (water 86.1%; ethanol 55.8%; methanol 113% methanol)	[36]
*Punica granatum* L.	pomegranate seed	TP, kaempferol -3-O-rutinoside	antioxidant (DPPH, ABTS)	80–280 °C, 5–120 min, 1:10–1:50 g/mL, 6.0 MPa	HPLC-DAD, UV, HPLC-ABTS^+^	leaching (water 40.6%; methanol 79.7%; ethanol 41.7%; acetone 45.5%), UAE (water11.3%; methanol 20.6%; ethanol 18.9%; acetone 15.2%), Soxhlet (methanol 71.4%; acetone 39.7%)	[144]
*Teucrium montanum* L.	aerial parts	rutin, naringin, epicatechin, etc.	antioxidant (DPPH, FRAP)	60–200 °C, 30 min, 1–10 MPa, 1:10 g/mL	HPLC-PDA, UV		[145]
*Paeonia lactiflora*	root	albiflorin, paeoniflorin		100–260 °C, 10–60 min,10–40 mL/g	HPLC	reflux (water 83.5%), UAE (ethanol 77.8%)	[146]
*Morus nigra* L.	fruits	TP, TF, cyanidin 3-glucoside, etc.	40–80 °C, 20–60 min, 2–6 mL/min, 15 MPa	tyrosinase inhibitory activity	UPLC-DAD-ESI-MS/MS	shaker (ethanol:water 116%), UAE (ethanol:water:TFA 134%)	[147]
*Stevia rebaudiana*	leaves	TP, stevioside, rebaudioside A	antioxidants (DPPH)	100–150°C, 30–60 min, 23 MPa, 1:10 g/mL	HPLC-UV, UV		[148]
*Erigeron breviscapu*s	whole parts	scutellarin, 20 inorganic elements, etc.	antioxidant (DPPH)	120–140 °C, 5–15 min, 150–420 um	HPLC, HPLC-MS	reflux (methanol 86.1%; ethanol 84.8%)	[149]
*Mangifera indica* L.	leaves	quercetin3-d-glucoside, mangiferin	antioxidant (DPPH)	100 °C, 4 MPa, 10 g/min, 3 h	UV, HPLC	SCCO_2_ (20% methanol 18.7%)	[150]
*Crocus sativus* L.	stigmas	picrocrocin, safranal, crocin		5–15 min, 105–125 °C	GC-MS, UV, HPLC		[151]
*Glycyrrhiza uralensis* Fisch	licorice root	TP, glycyrrhetic acid, glycyrrhizin, liquiritin	antioxidant (DPPH, reducing power)	50–300 °C, 10–60 min, 0.002–5 MPa	HPLC, UV-Vis		[152]

**Table 5 molecules-26-04004-t005:** Subcritical water extraction of carbohydrates.

Samples	Medicinal Parts	Compounds Extracted	Activity/Mixtures	Extraction Conditions	Analytical Methods	Other Extraction Methods (Solvent, Ratios of Yields)	Ref.
*Lycium barbarum*	berries	total sugar content	antioxidant (FRAP, TEAC), immunomodulatory	1:30 g/mL, 110 °C, 5 MPa	HPGPC	HWE (water 71.5%), UAE (water 89.9%), UWE (water 132%)	[11]
sunflower	sunflower heads	galacturonic acid, pectin		10–50 min, 2–8 mL/g, 100–140 °C, 0.2–1 MPa	TG/TGA, DSC, UV−vis, FTIR, HPSEC, NMR		[40]
*Aronia melanocarpa*	chokeberry stems	1 amino acid, 8 alcohols, 11 sugars, 2 fatty acids, etc.	antioxidant (DPPH), enzyme inhibitory activity	130 °C, 3.5 MPa, 20 min, 1:20 g/mL	GC-MS		[49]
*Lentinus edodes*	fruit bodies	hetero–polysaccharides, xylose, mannose, etc.	antioxidant (OH·, DPPH, ABTS)	120–160 °C,30–50 min, 0.033–0.05 g/mL	UV-vis, SEM, GC, GPC, FT-IR		[54]
*Lentinus edodes*	fruit bodies	l-rhamnose, d-arabinose, d-xylose, d-mannose	antioxidant (ABTS), growth inhibitory effect	100–150 °C, 10–30 min, 5 MPa	FT-IR, UV-Vis, AFM, GC, HP SEC-MALLS		[55]
*Lentinus edodes*	fruit bodies	polysaccharides, rhamnose, arabinose, xylose, etc.	antioxidant (DPPH, reducing power)	140 °C, 40 min, 1:25 g/mL, 5 MPa	GC, FT-IR, AFM, SEM		[56]
*Lentinula edodes*	fruit bodies	TCC, total β-glucan, chitin	HMGCR, immuno- modulatory	200 °C, 11.7 MPa, 15–60 min	GC-MS, HPSEC, NMR	UAE (water 65.2%), HWE (water 32.3%), SPE (water 33.0%)	[57]
*Grifola frondosa*	fruit bodies	total polysaccharide, total protein	antioxidant (DPPH, reducing power)	100–230 °C, 2–4 min, 20–100 mesh, 5 MPa	FT-IR, SEM	HWE (water ~87.8%)	[58]
*Sagittaria sagittifolia* L.	fruit bodies	polysaccharides	antioxidant (DPPH, ABTS, reducing power)	150–190°C, 12–20 min, 1:20–1:40 g/mL, pH 7–9	FT-IR, 1H and 13C NMR, UV	HWE (water 55.8%)	[59]
*Sagittaria sagittifolia* L.	fruit bodies	l-rhamnose, d-arabinose, d-xylose, d-mannose	antioxidant, immuno-modulatory	170°C, 16 min	HPLC, GC, SEM, IR, AFM, HPSEC-MALLS	HWE (water 75.6%); UAE (water 96.1%)	[60]
*Sagittaria sagittifolia* L.	fruit bodies	α-pyranose polysaccharide, β-pyranose polysaccharide	immuno-stimulatory	1 MPa, pH 7,170 °C, 16 min, 30:1 mL/g	IR, GC-FID, UV, HPSEC, AFM		[61]
*Cordyceps militaris*	fruit bodies	total sugars, protein and uronic acid		180 °C, 13 min, pH = 8, 21 mL/g	IR, GC, AFM, GPC-MALLS		[62]
wheat	bran	monosaccharide, etc.	antioxidants (DPPH)	160–180°C, 5–60 min	HPAEC-PAD, SEC		[153]
*Saccharina japonica*		fucoidan, fucose, glucose, galactose, mannose, etc.	antioxidant, antimitotic anti-proliferative	100–180 °C, 5–15 min, 2–8 MPa	FTIR, TGA, UV-Vis	convention (0.05 M HCl 100%)	[154]
*Citrus grandis* L.	pomelo peel	pectin		90–120 °C, 3–10 MPa	HPSEC-MALLS		[155]
*Theobroma cacao* L.	cacao pod husks	xylose, arabinose, etc.		121 °C, 30 min, 10.3 MPa	FT-IR, GC-FID, SEM	convention (4% citric acid 76.1%)	[156]
*Kappaphycus alvarezii. A*		ĸ-carrageenan, glucose, 3,6-anhydrogalactose, etc.	antioxidant (DPPH, ABTS)	60–180°C, 5 MPa, 5 min	FTIR, TGA, XRD	convention (water 94.3%; water with IL 101%)	[157]
*Pseuderanthemum palatiferum*	leaves	TCC, monosaccharides	anticoagulant, antioxidant	150–200°C, 5–10 mL/min	HPLC, GPC, NMR, UV	convention (0.1 M NaOH 48.8%)	[158]
wheat	bran	TCC, reducing sugar, arabinose, xylose, etc.	antioxidant, α-amylase inhibitory	140 °C, 5 MPa, 30 min	SEC-MALLS, FT-IR, DLS, DSC, UV	SBWE (water with citric acid 97.6%); UWE (water with citric acid 103%)	[159]
*Lycium barbarum* L.	fruits	polysaccharides	antioxidant (O2·, OH·, DPPH)	5 MPa, 25 mL/g, 110 °C, 1 h	UV	HWE (water 86.2%); UAE (water74.9%); UWE (water 111%)	[160]
*Cocos nucifera* L.	defatted coconut	mannose, galactosamine, xylose, rhamnose, etc.	antioxidant, hypoglycaemic, adsorption	1:10–1:50 g/mL, 10–50 min, 120–200 °C, 20–100 mesh	HPLC, XRD, TGA, DTGA, SEM, FT-IR		[161]
okara		polysaccharides, TP, TF	antioxidant (ABTS, DPPH)	1:30 g/mL, 160–230 °C, 10 min	UV		[162]
*Saccharina japonica*		polysaccharide, fucoidan, alginate	antioxidant (ABTS, DPPH, FRAP)	100 –150 °C, 1–5 MPa, 1:30–1:50 g/mL	IR, DSC, TGA, ^1^HNMR, HPLC, HPSEC-ELSD		[163]
*Passiflora edulis*	fruit peel	pectic polysaccharide, mannose, glucose, etc.	antioxidant (DPPH)	100–160 °C, 5.64–7.94 min, 10–30% ethanol	HPLC, UV, viscometer		[164]
*Chlorella vulgaris, Sargassum vulgare, Sargassum muticum, Porphyra* spp., *Cystoseira abies–marina, Undaria pinnatifida and Halopitys incurvus, Rosmarinus officinalis* L., *Thymus vulgaris, Verbena officinalis*	microalgae, algae, leaves	sugar, TP, melanoidins	antioxidant (ABTS, O_2_¯)	100–200 °C, 20 min, 10 MPa	UV		[165]
rice bran	bran	protein, TCC, TP	antioxidant (DPPH)	120–250 °C,0.5–5 mL/min	UV, UV-Vis		[166]
*Nizamuddinia zanardinii*		TCC, rhamnose, xylose, arabinose, fucoidan, fucose	antioxidant, anticancer, macrophage, etc.	425 rpm, 10–30 min, 90–150 °C, 0–40 mL/g, 0.75 MPa, 1500 W	FT-IR, GC-MS, SEM, UV, HPSEC-MALLS-RI		[167]
*Dendrobiumnobile* Lindl.	stems	polysaccharide, arabinose, galactose, glucose, etc.	antioxidant (OH·, ABTS)	0.5–1.5 MPa, 5–20 min 120–160 °C, 1:25 g/mL	UV−vis, GPC, HPLC, HPAEC		[168]
*Ecklonia maxima*		TP, polysaccharide, sulphate, and alginate	antioxidant (ABTS)	100–180 °C, 5–30 min, 10–50 mL/g, 4 MPa	UV, elemental analysis, ICP-MS	convention (70% ethanol 0%; 0.05 M HCl 20.1%)	[169]
*Vitis vinifera*	grape pomace	glucose, fructose, galactose, arabinose, mannose, etc.	antimicrobial, antioxidant (DPPH)	170–210 °C, 10 MPa, 5–10 mL/min	HPLC, UV		[170]
green coffee beans	spent coffee grounds	carbohydrates, phenolics	antioxidant, antibacterial	150–220 °C, 7 MPa, 10 mL/min,	HPLC, UV,		[171]
*Tamarindus indica*	seed	TP, xyloglucan	antioxidant (DPPH)	100–200°C, 1:20 g/mL	SEC, UV	convention (water 74.6%)	[172]
*Mentha arvensis*	leaves	carbohydrates, apocynin	antioxidant (DPPH)	180–260 °C, 1:20 g/mL, 5 min	HPLC, GC-MS, UV,		[173]

**Table 6 molecules-26-04004-t006:** SBWE of essential oils, alkaloids, quinones, terpenes, lignans, and steroids.

Samples	Medicinal Parts	Compounds Extracted	Extraction Conditions	Methods	Other Extraction Methods (Solvent, Ratios of Yields)	Ref.
**Essential oils**						
*Thymbra spicata* L.	leaves	α-thujene, α-pinene, terpinen-4-ol, p-cymene, γ-terpinene, 1-carvone, thymol, carvacrol, etc.	100–175 °C, 1–3 mL/min, 2–9 MPa, 30 min	GC-TOF/MS, GC-FID		[12]
*Aquilaria malaccensis*	leaves	butanal, cyclopentanone, acetoxyacetone, benzaldehyde, acetophenone, creosol, etc.	100–271 °C, 1–34 min, 0.08–0.22 g/mL	GC-MS, SEM, FT-IR	HD (95.4%)	[22]
Mentha piperita L.	peppermint leaves	TP, menthone, menthol, eriocitrin, etc.	40–160 °C, 10.3 MPa, 1–30 min	GC-MS, FID, HPLC	convention (methanol 53.2%)	[28]
*Coriandrum sativum* L.	coriander seeds	thujene, sabinene, pinene, myrcene, cymene, limonene, ocimene, terpinene, terpinolene, etc.	100–175 °C, 1–4 mL/min, 0.25–1 mm, 2 MPa, 20 min	GC-FID, GC-MS	HD (1.54-fold), Soxhlet (hexane 1.4-fold)	[174]
*Coriandrum sativum* L.	coriander seeds	3,4-dimethoxycinnamic acid, coumaric acid, sinapic acid, cis-and trans-linalooloxides, linalool, etc.	100–200°C, 10–30 min, 3–9 MPa	HPLC-MS/MS, GC-MS		[175]
*Kaempferia galangal* L.	rhizome	ethyl-p-methoxycinnamate, d-limonene, eucalyptol, tridecane, camphor, borneol, tetradecane, etc.	120 °C, 10 MPa, 30 min	GC-MS	HD (82.3%), UWE (100%)	[176]
*Piper betle*	leaves	4-allyl resorcinol, chavibetol	2 MPa, 10–90 min, 50–250 °C, 0.25–1 mm, 1–4 mL/min	HPLC-UV	convention (water 92.2–111%; methanol 96.6–110%)	[177]
*Aquilaria malaccensis*	leaves	nonacosane, triacontane, pentadecanal, 9-octadecenal, (Z)-, tetradecanal, tetrapentacontane, guaiacol	100–271 °C, 1–34 min	GC/MS, SEM, BET		[178]
laurel	leaves	α-phellandrene, β-pinene, 1,8-cineole, borneol, nona-3,7-dienol, isobornyl acetate, γ-terpineol, etc.	15 min, 50–200 °C, 1.5–15 MPa, 0.5–5.0 mL/min	GC-MS, GC-FID		[179]
*Citrus hystrix*	leaves	linalool, isopulegol, neoisopulegol, citronellal, 4-terpineol, citronellol, geraniol, menthoglycol, etc.	120–180 °C, 5–20 g/mL, 5–30 min	GC-MS	HD (28.2%)	[180]
*Coriandrum sativum* L.	coriander seeds	α-pinene, β-pinene, camphor, methylchavicol, γ-terpinene, linalool, geraniol, carvacrol, etc.	100–200 °C, 1:10 g/mL, 2 MPa, 20 min	GC-MS, GC-FID	HD (27.0%), Soxhlet (DCM 6.5-fold), SCCO_2_ (4-fold)	[181]
*Lavandula* L.	lavender flowers	a-thujene, a-pinene, camphene, sabinene, pinene, myrcene, hexylacetate, terpinene, limonene, etc.	125 °C, 3 MPa, 30 min	GC-MS, FID	HD (1.2-fold), US-HD (1.3-fold), NaCl-HD (1.3-fold)	[182]
**Alkaloids**						
*Sophora flavescens* Ait.	root	cytisine, matrine, sophoridine, sophocarpine, oxymatrine	70–190 °C, 5–14 min, 4.0–13.8 MPa	CE	ASE (ethanol 78.1%)	[16]
black tea brick	leaves	theophylline, epicatechin gallate, caffeine, etc.	120–180 °C, 7–42 min, 6–18 mL/min	HPLC		[46]
*Symphytum officinale* L.	root	lycopsamine, echimidine, lasiocarpine, symviridine	60–120 °C, 40 min	HPLC, LC-MS, MS^n^	HRE (methanol 2.8-fold)	[183]
*hydrastis canadensis*	root	hydrastine, berberine	100–160 °C,1–10 MPa, 5–60 min, 0.5–1.5 mL/min	HPLC-DAD	reflux (methanol 90.8%), UAE (methanol 106%)	[184]
cocoa	shells	TP, theobromine, theophylline, caffeine, epicatechin, etc.	120–220 °C, 15–75 min, 1:10–1:30 g/mL	HPLC, UV		[185]
*Musaceae*, *Beta vulgaris*	peels	dopamine, total betacyanin, betaxanthin	150°C, 5 min, 3 MPa, 1:20 g/mL	HPLC, UV-Vis	infusion (100%), decoction (1.2-fold), maceration (97.4%), UAE (101%), MAE (50.3%)	[186]
*Coffeea arabica, C. arabica, C. canephora var. robusta, C. canephora var. robusta*	coffee silver skin	total sugar, reducing sugar, protein, TP, caffeine, HMF, etc.	180–270 °C, 10 min, 1.0–5.3 MPa	HPLC, UV	convention (0.1 M HCl 96.6%; 0.1 M NaOH 1.5-fold)	[187]
**Quinones**						
*Rheum tanguticum*	root	damnacanthal	33–67 min, 100–200°C, 1.4–4.6 mL/min	HPLC, NMR, HSCCC		[17]
*Garcinia mangostana* Linn	mangosteen pericarps	TP, xanthone	120–160 °C, 1–10 MPa, 5–60 min, 10–30% DES	UV-vis, FT-IR, SEM		[188]
*Phaleria macrocarpa*	mahkota dewa fruits	mangiferin	4.0 MPa, 5 h, 50–150 °C	HPLC		[189]
*Lithospermum erythrorhizon*	root	shikonin, acetylshikonin, β-dimethylacrylshikonin, etc.	40–60 mesh, 120 °C, 5 MPa	UV, HPLC-ELSD	SCCO_2_ (86.3%), Soxhlet (ethyl acetate 95.4%), UWE (1.4-fold)	[190]
*Morinda citrifolia*	root	alizarin	4 MPa, 150 and 220 °C, 1.6–4 mL/min	RP-HPLC-UV		[191]
*Morinda citrifolia*	root	1,2-dihydroxyanthraquinone, alizarin	110–220 °C, 2–6 mL/min	UV-Vis	ethanol (3 d)	[192]
*Morinda citrifolia*	root		4 MPa, 150–200 °C, 2–6 mL/min	UV-Vis	convention (ethanol 81.16%), Soxhlet (ethanol 97.94%), UAE (ethanol 79.62%) SWBE (96.41%)	[193]
**Terpenes**						
*Hedyotis diffusa* Willd.	whole plants	ursolic acid	120–200 °C, 10–50 min, 20–40 mL/g, 0.6–3.0 MPa	HPLC-ESI-TOF-MS	maceration (ethanol 58.8%), HRE (ethanol 78.4%), UAE (ethanol 90.4%), MAE (ethanol 74.9%)	[13]
*Centella asiatica*	whole plants	asiatic acid, asiaticoside	100–250°C, 10–40 MPa, 5 h	HPLC, DLS		[14]
basil, oregano	leaves	limonene, citronellol, etc.	100 and 150 °C, 10 min	GC-FID		[48]
*Ganoderma lucidum*	fruits	ganodermanon-triol, ganoderic acids, lucidumol	100–200 °C, 5–10 MPa, 5–60 min	HPLC, SEC-UV, SEM, MALDI-TOF		[194]
*Orostachys japonicus*	stems, leaves	triterpene, camellia, etc.	110–260 °C, 5–20 min, 10 MPa	HPLC-MS		[195]
*Betula pendula*	birch bark	betulinic acid	160–200 °C, 10–30min, 10 MPa	HPLC		[196]
*Inula racemose*	plants	igalan, soalantolactone, alantolactone	23.2–56.8 min, 1.3–4.7 mL/min, 129.5–230.5 °C	HPLC, ^1^H-NMR ^13^C-NMR, MS	Soxhlet (ethanol 100%), UAE (ethanol 70.36%), SCCO_2_ (76.06%)	[197]
*Semen richonsanthis*	seeds	3,29-dibenzoylkarounidiol, polysaccharides	80–160 °C, 5.0–30.0 min	HPLC, UV, SEM		[198]
*Cucurbita pepo*	pumpkin peel	14 carotenoid compounds	120 °C, 3 h, 5 MPa	UV, HPLC	SCCO_2_ (75.4%)	[199]
*Betula pendula*	birch bark	sesquiterpenes, steroids	10 min, 100–200 °C	LC, GC/MS, NMR		[200]
*S. rebaudiana*	Bertoni leaves	steviol glycosides, tannins, chlorophyll A	100–160 °C, 5–10 min, 10.34 MPa, 1:3 g/mL	HPLC, UV, UV/Vis		[201]
**Lignans**						
*Linum usitatissimum* L.	flaxseed	SDG lignan, phenolics, flavonoids	160–180 °C, 5–60 min, 10 MPa	HPLC-MS/MS, UV		[41]
*Sesamum indicum* L.	sesame seeds	lignans, TP, flavonoids, flavonols	140–220 °C, 8–14 MPa, 0–95% ethanol, 0–75 min	UV		[42]
*Linum usitatissimum* L.	flaxseed	total fat content, SDG lignan	120–180 °C, 15–90 min, 10–13.8 MPa	HPLC-MS/MS, UV		[43]
*Sinopodophyllum hexandrum*	root	podophyllotoxin	12 mL/g, 3 MPa, 2ml/min, 120–240 °C	HPLC		[202]
**S** **teroids**						
*Pfaffia glomerata*, Amaranthaceae	ginseng root	sugar, fructooligosaccharides, beta-ecdysone	80–180 °C, 5–15 min, 2–12 MPa	HPLC-ELSD, HPLC		[24]
*Panax ginseng C.A. Meyer*	ginseng root	TP, maltol, panaxadiol, panaxatriol	150–200 °C, 5–30 min, 100 MPa	HPLC, UV	convention (water 32.6%; methanol 24.1%; ethanol 18.7%)	[25]
*Panax ginseng* C.A. Meyer	ginseng root	total ginsenosides, total sugar, 1-oleanane ginsenosides, etc.	120–200 °C, 20 min, 1:20 g/mL, 6.0 MPa	FT-IR, UV, UFLC-MS/MS	heating (water, 30.9%; ethanol 94.4%)	[26]
grapevine	root, wood, cane	E-piceid, E-piceatannol, E-resveratrol, E-parthenocissin, etc.	100–190 °C, 5–30 min, 10 MPa	LC-DAD/ESI-IT, Q-TOF, NMR	ASE (116% for cane; 103% for wood; 1.5-fold for root)	[203]
*Withania somnifera* L.	root leaves	TP, withanoside IV V, withaferin A, withanolide A, B	100–200 °C, 10–30 min, 10 MPa	HPLC, UV	maceration (water 31.7%), Soxhlet (ethanol 39.2%), MAE (methanol 45.8%)	[204]
*Acanthophyllum glandulosum*	root	saponin	121 °C, 0.15MPa, 15 min, pH 4–9	FT-IR, UV-vis, HPLC		[205]
*Vaccaria segetalis*	cowcock seed	vaccarosides, segetosides	125–175 °C, 15–180 min		USE (methanol 46.8%; water 27.9%; ethanol 5.2%)	[206]

## Data Availability

Data sharing not applicable. No new data were created or analyzed in this study.

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
