# Peer review of "Subcritical Water Extraction of Natural Products"

_molecules, 2021, doi:10.3390/molecules26134004_

Round 1
Reviewer 1 Report
This Review is focused on extraction of different natural compounds. Despite there is an impressive number of studies cited in this work, I cannot recommend accepting the Review in this form.
The work is not concise enough. The first part of the Review (Section 2) is discussed poorly and my recommendation is to rewrite this entire section and add more information.
Furthermore, I recommend including a more through explanation of mechanisms of extraction of different compounds. Also, very diverse groups of compounds are presented but not in enough detail.
Additionally, my recommendation is to include the drawbacks and/or challenges regarding subcritical water extraction. For example: a lot of studies discuss the formation of new antioxidants at highest temperatures, but also the formation of hydroxymethylfurfural.
Please include future perspective and possible development for this kind of extraction. In addition, subcritical water coupled with other green technologies is mentioned, but it might be discuss in a separate section.
Additional suggestions are as follows:
Please rephrase informal language. Example: “most eye-catching one”
Since the subcritical water extraction abbreviation is mentioned at the beginning of the work, it should be used throughout the entire Review.
Herbs used for Figure 1 refer to aromatic and medicinal plants? Tea leaves are also medicinal herbs, therefore, please clarify this. And I am not sure based on which data Figure 1 was prepared?
Please use Italic font for all Latin names
Please use uniform units: degree Celsius or K
“In addition, no waste disposal is required after subcritical water extraction, and thus, SBWE is also economical.” Could the authors please explain this statement? After subcritical water extraction filtration is usually applied and solid waste after filtration is generated.
I am not sure why curcuminoids are separated. Aren’t curcuminoids also polyphenol compounds? Additionally, they are poorly discussed and only two references are mentioned. There are more available published works with subcritical water extraction of curcuminoids on Google Scholar.
Also, flavonoids belong to polyphenols. Since they are widely investigated, they can be presented separately from phenols. But at least one after another.
Citing references in text is not consistent. For example, some citations have the Second name et al. form, and others use the First name et al. or First and Second name et al. form.
Non-consistency: medicinal plants, ethnomedicines, Tibetan medicine, etc.
Latin names of materials should be written in the Tables.
“The optimal temperature ranges from 120 to 200 °C for extracting the natural products mentioned above” is not true since it is stated, “…optimal extraction temperature of organic acids is mainly between 130-240 °C”.
Author Response
Response to reviewer comments
Reviewer 1
The work is not concise enough. The first part of the Review (Section 2) is discussed poorly and my recommendation is to rewrite this entire section and add more information.
As requested, we have reorganized and rewritten Section 2.
Furthermore, I recommend including a more through explanation of mechanisms of extraction of different compounds. Also, very diverse groups of compounds are presented but not in enough detail.
We have added one subsection to discuss the extraction mechanisms of SBWE in the revised manuscript. We have also added additional figures to demonstrate how SBWE works.
Additionally, my recommendation is to include the drawbacks and/or challenges regarding subcritical water extraction. For example: a lot of studies discuss the formation of new antioxidants at highest temperatures, but also the formation of hydroxymethylfurfural.
We fully agree. Potential analyte degradation during SBWE at higher temperatures needs to be addressed. The corresponding author’s research group has published several studies investigating the stability and degradation of organics under subcritical water conditions. If one plans to conduct SBWE, he/she must evaluate the potential decomposition of analytes in the study. We have added this point in the revised manuscript.
Please include future perspective and possible development for this kind of extraction. In addition, subcritical water coupled with other green technologies is mentioned, but it might be discuss in a separate section.
SBWE future perspectives are added to the revision.
We have inserted a subsection to address SBWE coupling with liquid chromatography to deepen readers’ understanding of this useful coupling technology.
Additional suggestions are as follows:
Please rephrase informal language. Example: “most eye-catching one”
We have fixed this and other similar informal language.
Since the subcritical water extraction abbreviation is mentioned at the beginning of the work, it should be used.
We have made such changes in most cases.
Herbs used for Figure 1 refer to aromatic and medicinal plants? Tea leaves are also medicinal herbs, therefore, please clarify this. And I am not sure based on which data Figure 1 was prepared?
Sources of natural products include a large number of plants, bacteria, fungi, insects, arachnids, marine organisms, and higher-order animals (National center for complementary and integrative health). We followed the US National Center for Complementary and Integrative Health’s classification in the revised manuscript. The word herb used in this paper includes both seasoning (aromatic) and medicinal plants. We reorganized and revised section 2 to comply with the National Center for Complementary and Integrative Health’s classification.
Please use Italic font for all Latin names
Fixed as requested.
Please use uniform units: degree Celsius or K
All pressure units are changed to MPa and temperature to degree Celsius.
“In addition, no waste disposal is required after subcritical water extraction, and thus, SBWE is also economical.” Could the authors please explain this statement? After subcritical water extraction filtration is usually applied and solid waste after filtration is generated.
No waste disposal refers to the liquid wastes generated after extractions involving organic solvents. Since water is not toxic, and thus, water extractants do not require waste disposal for extraction of natural products. The solid wastes generated by SBWE of plants also do not require waste disposal for plants containing no toxic species.
I am not sure why curcuminoids are separated. Aren’t curcuminoids also polyphenol compounds? Additionally, they are poorly discussed and only two references are mentioned. There are more available published works with subcritical water extraction of curcuminoids on Google Scholar.
We agree with this reviewer and reclassified them as polyphenols.
Also, flavonoids belong to polyphenols. Since they are widely investigated, they can be presented separately from phenols. But at least one after another.
We moved polyphenols to section 4.2.
Citing references in text is not consistent. For example, some citations have the Second name et al. form, and others use the First name et al. or First and Second name et al. form.
Fixed
Non-consistency: medicinal plants, ethnomedicines, Tibetan medicine, etc.
Fixed
Latin names of materials should be written in the Tables.
Latin names of materials have been written in the Tables.
“The optimal temperature ranges from 120 to 200 °C for extracting the natural products mentioned above” is not true since it is stated, “…optimal extraction temperature of organic acids is mainly between 130-240 °C”.
The sentence “The optimal temperature ranges from 120 to 200 °C for extracting the natural products” has been revised to “The optimal temperature ranges from 130 to 240 °C for extracting the natural products”.
Reviewer 2 Report
Subcritical Water Extraction of Natural Products
This review focuses on subcritical water extraction of natural products. The authors have set the stage by declaring a similar review done by Sefater Gbashi and co-workers.
The review is very well written and provided a new insight of the subcritical water extraction of natural product. There is a lack of review on this area and this review fits well into extant literature. This is because the last reported review discussed the principles, mechanisms, static and dynamic modes, and factors affecting subcritical water extraction efficiency.
I have some minor comment:
Abstract: Pls improve on the flow of the argument of the review: especially on what is the current state this technology.
INTRODUCTION:
For thousands of years, herbal medicine has played a vital role in treating diseases, especially in East Asia. The bioactive components in herbs and plants are the basis to prevention and treatment of many diseases [1,2]. Due to its relatively low side effects against chemically synthesized drugs, much attention is given to the extraction and separation of a wide range of bioactive compounds from herbs and plants. The thousand-year old extraction process of the active pharmaceutical ingredients (APIs) from medicinal herbs is the traditional herbal decoction (THD) method. However, there are some defects in THD, such as a long extraction time and decomposition of active pharmaceutical ingredients. Methanol, ethanol, n-hexane, petroleum ether, diethyl ether, chloroform, ethyl acetate, and glycerol are often used as extraction solvents to increase the extraction efficiency and reduce extraction time [3]. Obviously, these organic solvents are volatile, flammable, mostly toxic, and expensive. Thus, they are not safe extraction fluids for herbs, plants, fruits, vegetables, and the like.
The first part of the Introduction is well organised, however the cited references 1 and 2 seems not appropriate:
- Gowd, V.; Karim, N.; Shishir, M.R. I.; Xie, L.; Chen, W. Dietary polyphenols to combat the metabolic diseases via altering gut microbiota. Trends Food Sci. Tech. 2019, 93, 81–93.
- Sedem, D.C.; Duan, Y.; Zhang, H.; Golly, M.K.; Ma, H. Enhanced screening of key ultrasonication parameters: total phenol content and antioxidant activity assessment of Tartary buckwheat water extract. Sep. Sci. Technol. 2019, 55, 3242–3251.
Based on the listed subheading, the text aligned well with the aims of the study:
-extraction of flavonoids, carbohydrates, glycosides, organic acids, polyphenolics, alkaloids, essential oils, quinones, terpenes, lignans, steroids and curcumins
- overview on subcritical water extraction conditions, the function and activities of the active ingredients and the subcritical extracts, analysis methods, and comparison with other extraction methods for the above mentioned 12 classes of natural products.
- Sample Matrices Extracted by Subcritical Water
2.1. Herbs
2.2. By-Products
2.3. Vegetables
2.4. Algaes
2.5. Fruits
2.6. Oilseed Crops
2.7. Shrubs, Grains and Tea Leaves
- SBWE Systems
3.1. Development of SBWE Systems
3.2. Modes of SBWE
- Compounds Extracted by Subcritical Water
4.1. Flavonoids
4.2. Organic Acids
4.3. Polyphenols
4.4. Glycosides
4.5. Carbohydrates
4.6. Essential Oils
4.7. Alkaloids
4.8. Quinones
4.9. Terpenes
4.10. Lignans
4.11. Steroids
4.12. Curcumins
Comments of Figures and Tables
Figure 3-8: Optimum extraction conditions for several substances should be presented as one figure or next to each other to facilitate reader’s understanding
Figure 3. Optimum extraction conditions for flavonoids.
Figure 4. Optimum extraction conditions for organic acids.
Figure 5. Optimum extraction conditions for polyphenols and glycosides.
Figure 6. Optimum extraction conditions for carbohydrates, essential oils and alkaloids.
Figure 7. Optimum extraction conditions for quinones, terpenes and lignans.
Figure 8. Optimum extraction conditions for steroids and curcumins.
Suggest to combine Table 1-12 Subcritical water extraction of substances together into 1 table
Table 1. Subcritical water extraction of flavonoids.
Table 2. Subcritical water extraction of organic acids
Table 3. Subcritical water extraction of polyphenols.
Table 4. Subcritical water extraction of glycosides.
Table 5. Subcritical water extraction of carbohydrates.
Table 6. Subcritical water extraction of essential oils.
Table 7. Subcritical water extraction of alkaloids.
Table 8. Subcritical water extraction of quinones.
Table 9. Subcritical water extraction of terpenes.
Table 10. Subcritical water extraction of lignans.
Table 11. Subcritical water extraction of steroids.
Table 12. Subcritical water extraction of curcumins.
- Conclusions : the conclusion summarised well the essences of the review
Author Response
Response to reviewer comments
Reviewer 2
This review focuses on subcritical water extraction of natural products. The authors have set the stage by declaring a similar review done by Sefater Gbashi and co-workers.
The review is very well written and provided a new insight of the subcritical water extraction of natural product. There is a lack of review on this area and this review fits well into extant literature. This is because the last reported review discussed the principles, mechanisms, static and dynamic modes, and factors affecting subcritical water extraction efficiency.
We thank the reviewer for this nice comment!
I have some minor comment:
Abstract: Pls improve on the flow of the argument of the review: especially on what is the argument of the review:.
We have revised the abstract to enhance the need for and the argument of this review. We also added the disadvantages and challenges of the SBWE technique.
INTRODUCTION:
For thousands of years, herbal medicine has played a vital role in treating diseases, especially in East Asia. The bioactive components in herbs and plants are the basis to prevention and treatment of many diseases [1,2]. Due to its relatively low side effects against chemically synthesized drugs, much attention is given to the extraction and separation of a wide range of bioactive compounds from herbs and plants. The thousand-year old extraction process of the active pharmaceutical ingredients (APIs) from medicinal herbs i the traditional herbal decoction (THD) method. However, there are some defects in THD, such as a long extraction time and decomposition of active pharmaceutical ingredients. Methanol, ethanol, n-hexane, petroleum ether, diethyl ether, chloroform, ethyl acetate, and glycerol are often used as extraction solvents to increase the extraction efficiency and reduce extraction time [3]. Obviously, these organic solvents are volatile, flammable, mostly toxic, and expensive. Thus, they are not safe extraction fluids for herbs, plants, fruits, vegetables, and the like.
The first part of the Introduction is well organised, however the cited references 1 and 2 seems not appropriate:
- Gowd, V.; Karim, N.; Shishir, M.R. I.; Xie, L.; Chen, W. Dietary polyphenols to combat the metabolic diseases via altering gut microbiota. Trends Food Sci. Tech. 2019, 93, 81–93.
- Sedem, D.C.; Duan, Y.; Zhang, H.; Golly, M.K.; Ma, H. Enhanced screening of key ultrasonication parameters: total phenol content and antioxidant activity assessment of Tartary buckwheat water extract. Sep. Sci. Technol. 2019, 55, 3242–3251.
Thank you for this suggestion. The originally cited reference 1 has been replaced (see attached directly below) and reference 2 has been deleted in the revision.
- Colegate, S.M.; Molyneux, R.J. Bioactive natural products, 1993 (eds) CRC Press, Boca Raton.
Based on the listed subheading, the text aligned well with the aims of the study:
-extraction of flavonoids, carbohydrates, glycosides, organic acids, polyphenolics, alkaloids, essential oils, quinones, terpenes, lignans, steroids and curcumins
- overview on subcritical water extraction conditions, the function and activities of the active ingredients and the subcritical extracts, analysis methods, and comparison with other extraction methods for the above mentioned 12 classes of natural products.
- Sample Matrices Extracted by Subcritical Water
2.1. Herbs
2.2. By-Products
2.3. Vegetables
2.4. Algaes
2.5. Fruits
2.6. Oilseed Crops
2.7. Shrubs, Grains and Tea Leaves
- SBWE Systems
3.1. Development of SBWE Systems
3.2. Modes of SBWE
- Compounds Extracted by Subcritical Water
4.1. Flavonoids
4.2. Organic Acids
4.3. Polyphenols
4.4. Glycosides
4.5. Carbohydrates
4.6. Essential Oils
4.7. Alkaloids
4.8. Quinones
4.9. Terpenes
4.10. Lignans
4.11. Steroids
4.12. Curcumins
Comments of Figures and Tables
Figure 3-8: Optimum extraction conditions for several substances should be presented as one figure or next to each other to facilitate reader’s understanding
Figure 3. Optimum extraction conditions for flavonoids.
Figure 4. Optimum extraction conditions for organic acids.
Figure 5. Optimum extraction conditions for polyphenols and glycosides.
Figure 6. Optimum extraction conditions for carbohydrates, essential oils and alkaloids.
Figure 7. Optimum extraction conditions for quinones, terpenes and lignans.
Figure 8. Optimum extraction conditions for steroids and curcumins.
As requested, figures are either combined presented next to each other in the revised manuscript.
Suggest to combine Table 1-12 Subcritical water extraction of substances together into 1 table
Table 1. Subcritical water extraction of flavonoids.
Table 2. Subcritical water extraction of organic acids
Table 3. Subcritical water extraction of polyphenols.
Table 4. Subcritical water extraction of glycosides.
Table 5. Subcritical water extraction of carbohydrates.
Table 6. Subcritical water extraction of essential oils.
Table 7. Subcritical water extraction of alkaloids.
Table 8. Subcritical water extraction of quinones.
Table 9. Subcritical water extraction of terpenes.
Table 10. Subcritical water extraction of lignans.
Table 11. Subcritical water extraction of steroids.
Table 12. Subcritical water extraction of curcumins.
While it is a good idea to combine tables, presenting all information in one table will overwhelm readers in finding the information in the table while reading the text. They have to go back and forth between the table and the text. Since Tables 1-5 are larger tables by each self, we combined the relatively smaller Tables 6-12. into one table.
- Conclusions : the conclusion summarized well the essences of the review.
Thank you again.
Round 2
Reviewer 1 Report
The authors revised and corrected the Manuscript according to most of the suggestions. Therefore, I recommend accepting of the manuscript, after some additional corrections:
Please include references or describe based on which data Figure 2 (Types of sample matrices extracted by subcritical water) was prepared. Were the studies cited in present work or data provided by searching some scientific base?
Please use abbreviation for subcritical water extraction in the entire manuscript at the appropriate places (except in titles, figures, and tables captions). At most places the abbreviation is used, however, at others there is still the full name, such as:
Row 31 “Another advantage is that no liquid waste disposal is required after subcritical water extraction”
Row 40 “…analyte stability checks when carrying out subcritical water extractions”
Row 229 “…searchers have attempted to couple subcritical water extraction…”
Additionally, authors stated that Latin names of materials/samples are written in the Tables. However, that is not the case. Please include Latin names of all materials. Also, since at some places it is written just “pomegranate”, whereas at others the investigated plant part is also mentioned, such as “chestnut shells”, my recommendation is to include an additional column stating the investigated part of material. Common English name could also be included, but Latin name is necessary.
There are still many technical errors, therefore, revise the entire work carefully.
